# Consumers' risk perception, market demand, and firm innovation: Evidence from China

**Jing Cao**[1], **Haiwei Jiang**[2], **Xiaomeng Ren**[3]*, **Jinchuan Shi**[4]

**1** School of Economics, Zhejiang University, Zhejiang, China, **2** School of International Trade and Economics, Central University of Finance and Economics, Beijing, China, **3** Research Center for Regional Coordinated Development & China Academy of West Region Development, Zhejiang University, Zhejiang, China, **4** Center for Research of Private Economy, Zhejiang University, Zhejiang, China

* xiaomengren@zju.edu.cn

**Data Availability Statement:** All relevant data are within the manuscript and its Supporting Information files.

**Funding:** This study was supported by National Social Fund Youth project(No 19CJL052). The

## Abstract

Major product safety incidents often cause widespread concern among consumers, and these product safety incidents will stimulate consumers' psychology, change their risk perception, and affect the demand for products and services of risk consumers. The change in consumer demand will eventually lead to a change in firm innovation decisions. Using Chinese firm-level data, this paper employs the news reporting of the Bawang event as a quasi-natural experiment to study the impact of risk perception changes on innovation. The empirical results of this study show that increasing consumers' risk perception caused by the negative news coverage of defective products motivates firms to increase their innovation. The effects are heterogeneous, where firms with private ownership and in developed regions are more likely to increase innovation activities. This study suggests that the relationship between consumers' risk perception and firm innovation is primarily driven by market demand. Moreover, the positive effects of risk perception on innovation are more prominent for downstream firms and those having a smaller technological distance.

## 1. Introduction

In academic research, two widely accepted theories explain the motivation behind innovation activities: technology-induced theory and demand-induced theory. The technology-induced theory holds that innovative activities are driven by technological development and determined by the discovery of new knowledge, the R&D efficiency of researchers and institutions, and the cost required to promote new technologies [1,2]. On the other hand, the demand-induced theory holds that, like other economic activities, innovative activities are essential for profit and are affected and constrained by market demand [3–5]. Demand may steer technological progress through various mechanisms, including market size, heterogeneity of consumer demand, understanding of local needs, and feedback from customers and critical users [6–10]. Although the mechanism of demand-induced innovation has been discussed in some literature, only a few scholars have noticed that consumers' risk perception is a demand-side influencing factor.

funders had no role in study design, data collection and analysis.

**Competing interests:** The authors have declared that no competing interests exist.

Consumers' risk perception refers to the uncertainty that a consumer may feel when making a purchase decision. They may not be sure whether the purchase results will meet their requirements, and sometimes, the results may not be pleasant [11]. When a sudden accident occurs, such as a product harm crisis, negative information about the product and the firm involved spreads rapidly in the market. This causes widespread concern among consumers, increases their risk perception, and ultimately affects their purchase decisions [12–14]. This change in consumer behavior leads to a shift in market demand, affecting manufacturing enterprises' innovation activities [15,16].

Changes in consumers' risk perception can have a significant impact on firm innovation, and this impact can differ from other demand-induced forces. There are several reasons why this is so. Firstly, it is often difficult for consumers to have access to all the information they need about risks, and their understanding of product risks can be influenced by small probability events and media overexposure [17,18]. Secondly, the impact of increased risk perception on firm innovation is ambiguous. When consumers perceive more risk, their willingness to pay for security increases, and the demand for safer products increases. This, in turn, may increase firms' willingness to innovate products and technologies. However, on the other hand, the willingness to innovate on products and technologies that are considered high-risk may decline [19]. Thirdly, changes in risk perception also have an externality on firm innovation. Sudden accidents can impact not only the companies directly involved but also the firms and industries related [20,21]. Given these specificities of consumers' risk perception, research on the effect of consumers' risk perception on firm innovation can be a valuable addition to the existing literature on "demand-induced innovation".

This paper uses the relevant data of Chinese firms from 2008 to 2014, taking Chinese consumers, Chinese manufacturing firms, and the Chinese market as research objects, employs a difference-in-differences (DID) approach by using the Bawang event, a sudden accident, as a quasi-natural experiment to identify the impact of changes in consumers' risk perception on firm innovation. Our research shows that the rise in consumers' risk perception of products containing dioxane after the Bawang event led to a significant increase in patent applications in the daily chemical and pharmaceutical manufacturing sectors, which rose by approximately 7.6% compared to other manufacturing industries.

In 2010, Next Magazine published an article titled "Bawang Shampoo Causes Cancer". The article claimed that Bawang shampoo contained high levels of cancer-causing dioxane, and using the shampoo could be life-threatening. The report received widespread attention from society and was covered by mainstream media and websites. Consequently, consumers became more aware of the risks associated with similar products, which affected their purchasing decisions and demand. We use the Bawang event as exogenous shocks in the DID approach. The accidental nature of the Bawang event and the extensive media coverage at the time provide quite convincing evidence of the externality of our shock.

We have decided to utilize the data from 2008 to 2014 for two main reasons. Firstly, the Bawang event occurred in 2010, which makes it reasonable to use data from 2008 to 2014. This ensures the accuracy and effectiveness of estimation results following the method on samples. Secondly, a new drug approval reform in China in 2015 significantly impacted China's pharmaceutical innovation. Therefore, data from 2015 onwards is unsuitable for inclusion in our research sample.

The literature most directly related to our study is Galasso and Luo (2021) [22]. While we have borrowed their ideas and methods to overcome the limitations of previous research in this field, such as endogenous issues, some differences exist between our research and that of Galasso and Luo (2021) [22]. Firstly, compared with Galasso and Luo, we have provided a more detailed discussion on changes in consumers' perceived risk. We have a comprehensive

and detailed discussion on two aspects of consumers' perceived risk: social risk and functional risk. Secondly, this paper introduces economic concepts such as industrial chain, industrial correlation, and technology spillover effect into the research of "demand-driven innovation" within the framework of risk perception. Our research indicates that the influence of consumers' risk perception on firm innovation depends on their position in the industrial chain and technology spillover. We have two main findings: (1) Firms closer to the downstream of the industrial chain are more sensitive to changes in consumers' perceived risks, leading to a more significant impact on their innovation activities. (2) Cross-industry technology spillovers are more likely to occur between firms and industries with high technical similarity when a sudden accident occurs. The higher the technological similarity with the firm involved in the sudden accident, the more significant the impact of consumers' risk perception on the firm innovation. Thirdly, the study conducted by Galasso and Luo analyzes the course of technological progress using patent and FDA data at the patent classification level. Meanwhile, we choose firm-level data and analyze Chinese manufacturing firms as micro subjects of innovation activities. We focus on the firms' decisions when responding to sudden accidents to recover their reputation and compensate for the losses. We also add a discussion on firm heterogeneity to enhance the analysis.

The rest of this paper is as follows. Section 2 introduces the relevant literature and theoretical hypothesis of this study. Section 3 introduces the event background and changes in consumers' risk perception. Section 4 presents the data and identification method. Section 5 reports the analysis of the empirical results. Section 6 discusses the mechanisms. Section 7 makes further discussions. Section 8 provides conclusions and recommendations.

## 2. Literature and theoretical hypothesis

### 2.1. Consumers' risk perception and market demand

**2.1.1 Consumers' perceived risk.** The concept of perceived risk was first proposed by Bauer(1967) based on psychological research [11]. According to Bauer, consumers tend to anticipate the outcomes of their purchase decisions, and some of those outcomes may not be desirable. The uncertainty of the results of the purchase decision is what is referred to as perceived risk. When studying consumer behavior using the perceived risk theory, purchasing is viewed as risk-taking behavior because consumers cannot ascertain the outcome of using a product before purchasing it. Therefore, consumers bear a certain level of risk. This view has been supported by various scholars such as Derbaix (1983) and Cunningham et al. (2005) [23,24].

Generally, the changes of perceived risk are discussed from six aspects: (1) Financial risk: the value of the product does not match the cost; (2) Functional risk: the product cannot be used, or the function cannot achieve the expected effect; (3) Physical risk: the risk of harm to consumers when the product is poorly designed; (4) Psychological risk: When the product may be inconsistent with the consumer's self-image or the purchased product does not meet the expected level, the consumer has the risk of psychological or self-perceived harm. For example, the products' quality does not match, and doubts arise about their buying ability. (5) Social risk: the risk that others do not recognize the products consumers purchase. (6) Time risk: loss of time and effort when purchasing a product [25,26].

**2.1.2. Sudden accident, consumers' risk perception and market demand.** The product harm crisis is a typical type of sudden accident. The product harm crisis, as defined by Siomkos and Kurzbard (1994) [27], is when a product becomes defective or dangerous to consumers and receives widespread publicity. The market is flooded with negative information about the companies and products involved in such situations. This causes physical and

psychological harm to consumers, who experience negative emotions such as tension and fear. As a result, consumers' risk perception associated with the crisis products increases, and their willingness to buy and use such products decreases. This phenomenon has been extensively studied by researchers such as Rogers et al. (2007), Becker and Steven (2010), Fullerton et al. (2003), and Kotler et al. (2018) [28–31]. From an individual perspective, consumer demand for dangerous products decreases, and demand for better products increases. From a market-wide perspective, since market demand is the aggregate of individual needs, the overall market demand also changes accordingly [32].

**2.1.3. Consumer trust.** Our research has relevance to the theory of consumer trust despite not being directly involved. The concept of consumer trust is closely linked with perceived risk, which has been found by Jarvenpaa et al. (2000) and Corbitt et al. (2003) to significantly impact consumers' purchasing decisions [33,34], mainly by influencing their trust. Not only does consumer trust directly affect purchasing decisions, but it also profoundly impacts brand reputation, market share, and the long-term sustainable development of firms [35]. The trust consumers place in a firm to meet their expectations is known as consumer trust. This encompasses trust in the firm's ability to perform and trust in honesty and kindness [36,37]. When buyers have positive experiences, they are more likely to develop trust in the firm in the future. Trust is built through accumulating satisfying experiences over time [38]. Consumers who trust firms and their brands are more likely to spread positive word-of-mouth [39].

Previous studies on consumers' risk perception have primarily used theories and methods from psychology, management, and related fields. These studies focused on measuring risk perception and identifying the factors that affect it but did not utilize economic methods. Moreover, the research perspective of these studies is limited to individual consumers. In contrast, our paper uses economic methods to expand the research perspective to the entire market.

Although some scholars have studied how changes in consumers' risk perception and trust affect their purchasing decisions following a sudden accident, such as a product harm crisis, there is still a need for more comprehensive discussions on the strategies that firms should adopt to restore consumers' trust and reduce their risk perception. Previous studies mainly focused on external strategies, such as crisis public relations and advertising, and overlooked internal strategies, such as improving products and technology through innovation. Based on the above theoretical analysis, the following hypotheses are proposed:

**H1:** Sudden accident has an impact on consumers' risk perception.

**H2:** Consumers' risk perception has an impact on market demand.

## 2.2 Market demand and firm innovation

In discussing the causes of innovation activities, there are two widely accepted theories in academic research: technology-induced theory and demand-induced theory. For a long time, people believed that innovation was caused by the knowledge progress of scientists and technological inventions outside the economic system, which was called the "technology-induced theory" in Schumpeter's innovation theory. The technology-induced theory holds that innovative activities are driven by technological development and determined by the discovery of new knowledge, the R&D efficiency of researchers and institutions, and the cost required to promote new technologies [1,2]. However, since the 1960s, the "demand-induced theory" began to rise [3,40], and the profit motive of firms is considered to be the primary source of innovation; with the arrival of the third industrial revolution, all kinds of

technological changes reflect more demand-oriented, Acemoglu (2002) called it "inductive technological innovation" [4]. In the 21st century, the "demand-induced innovation" theory has been extended to the micro level. Foellmi and Zweimüller (2008) pointed out through a series of studies that the guiding effect of demand on innovation depends on the income distribution structure. As consumers' preferences are heterogeneous, firms will constantly innovate to meet the needs of different consumers. High-income consumers buy new products, and low-income consumers buy necessities. With the increase in overall income and the change in income distribution structure, the expansion of market scale, product diversification, and quality improvement of related products are promoted. That is, the production innovation of the firm is promoted [5].

Although the concept of "demand-induced innovation" has been widely recognized and verified by theoretical models, there are few empirical studies on it. Existing empirical studies mainly discuss the impact of market demand on technological innovation from the perspectives of market size, heterogeneity of consumer demand, understanding of local needs, and feedback from customers and critical users. Acemoglu and Linn (2004), Blume-Kohout and Sood (2013), and Dubois et al. (2015) explored the impact of market size on firm innovation activities [41–43]. They found that expanding the market size helps stimulate the R&D investment and innovation momentum of the firm, thus promoting technological progress and improving industrial competitiveness. The study of Adner and Levinthal (2001) focuses on the impact of the diversity of consumer demand on technological innovation. They point out that the heterogeneity of consumer demand encourages firms to develop diversified products to meet the needs of different consumer groups, thus promoting product innovation and market competition [6]. The research of Fabrizio and Thomas. (2011) emphasizes the importance of understanding local demand patterns for firm innovation. They pointed out that firms should have a deep understanding of the needs and characteristics of the local market in order to better position products and develop innovative solutions that meet the market needs [8]. Research by Hippel (1986) and Chatterji and Fabrizio (2012) highlight the importance of user participation in the innovation process [9,10]. They point out that feedback from customers and key users can help companies improve existing products, develop new ones, and better understand market needs.

This paper believes that there are still some problems worthy of further discussion. First, we can discuss how market demand drives innovation from a richer perspective. For example, changes in market demand caused by risk perception have received little empirical and theoretical attention. Second, the endogeneity problem has yet to be well solved in previous studies, and it is not easy to get an unbiased and consistent estimate. This paper holds that the most effective way to study the relationship between market demand and firm innovation is to find a reasonable exogenous impact and estimate it using the quasi-natural experiment method. Third, industrial heterogeneity and industry correlation have been neglected in previous studies. The same exogenous impact will have different effects on the innovation behavior of different industries. Even in the same industrial chain, the industry in the upstream and the industry in the downstream of the industrial chain will have different changes in innovation behavior when the market demand changes [44,45]. In addition, due to the inter-industry correlation and technology spillover effect, the change of innovation behavior in one industry will affect the innovation of other industries [46,47], which is also worth noting. Based on the above theoretical analysis, the following hypothesis is proposed:

**Hypothesis 3 (H3).** Market demand has an impact on firm innovation.

## 2.3 Consumers' risk perception and firm innovation

Much of the literature shows that demand can guide technological progress through various mechanisms. However, so far, changes in market demand driven by consumers' risk perception have received little empirical and theoretical attention. Changes in consumers' risk perception can have a significant impact on firm innovation, and this impact can differ from other demand-induced forces. There are several reasons why this is so. Firstly, it is often difficult for consumers to access all the information they need about risks, and their understanding of product risks can be influenced by small probability events and media overexposure [17,18]. Secondly, the impact of increased risk perception on firm innovation is ambiguous. When consumers perceive more risk, their willingness to pay for security increases, and the demand for safer products increases. This, in turn, may lead to an increase in the willingness of firms to innovate products and technologies. However, on the other hand, the willingness to innovate on products and technologies that are considered high-risk may decline [19]. Thirdly, changes in risk perception also have an externality on firm innovation. Sudden accidents can impact not only the firms directly involved but also the firms and industries related [20,21].

The literature most directly related to our study is Galasso and Luo (2021) [22]. Galasso and Luo (2021) claimed that over-radiation accidents and their media coverage have increased the awareness levels of both patients and medical providers about medical radiation risk. They used data on patents and FDA product clearances and difference-in-differences regression results to show that the increased perception of radiation risk spurred the development of new technologies that mitigated the risk and led to more new products. While we have borrowed their ideas and methods, there are some ways in which our research differs from that of Galasso and Luo.

Firstly, Galasso and Luo noted that over-radiation accidents and their extensive media coverage can lead to an increased perception of risk regarding medical radiation among patients and medical providers. However, they did not provide a detailed analysis of how the perception of risk changes over time. We have chosen to discuss two aspects of consumers' perceived risk, namely social risk and functional risk, and provide a more detailed explanation of the changes in consumers' perceived risk over time.

Secondly, this paper introduces economic concepts such as industrial chain, industrial correlation, and technology spillover effect into the research of "demand-driven innovation" within the framework of risk perception. This has yet to be done in previous studies. We have presented our two hypotheses and conducted experiments to verify them: (1) Consumers' risk perception impacts firm innovation, and the extent of the impact depends on the firm's position in the industrial chain. (2) Consumers' risk perception impacts firm innovation and the extent of the impact related to technology spillover.

Thirdly, the study conducted by Galasso and Luo analyzes the course of technological progress using patent and FDA data at the patent classification level. Meanwhile, we choose firm-level data and analyze Chinese manufacturing firms as micro subjects of innovation activities. We focus on the decisions made when responding to sudden accidents to recover their reputation and compensate for the losses. We also add a discussion on firm heterogeneity to enhance the analysis. Based on the above discussion, the following hypothesis is proposed:

**H4:** Consumers' risk perception has an impact on firm innovation.

**H5:** Consumers' risk perception has an impact on firm innovation, and the extent of the impact depends on the firm's position in the industrial chain.

**H6:** Consumers' risk perception has an impact on firm innovation and the extent of the impact related to technology spillover.

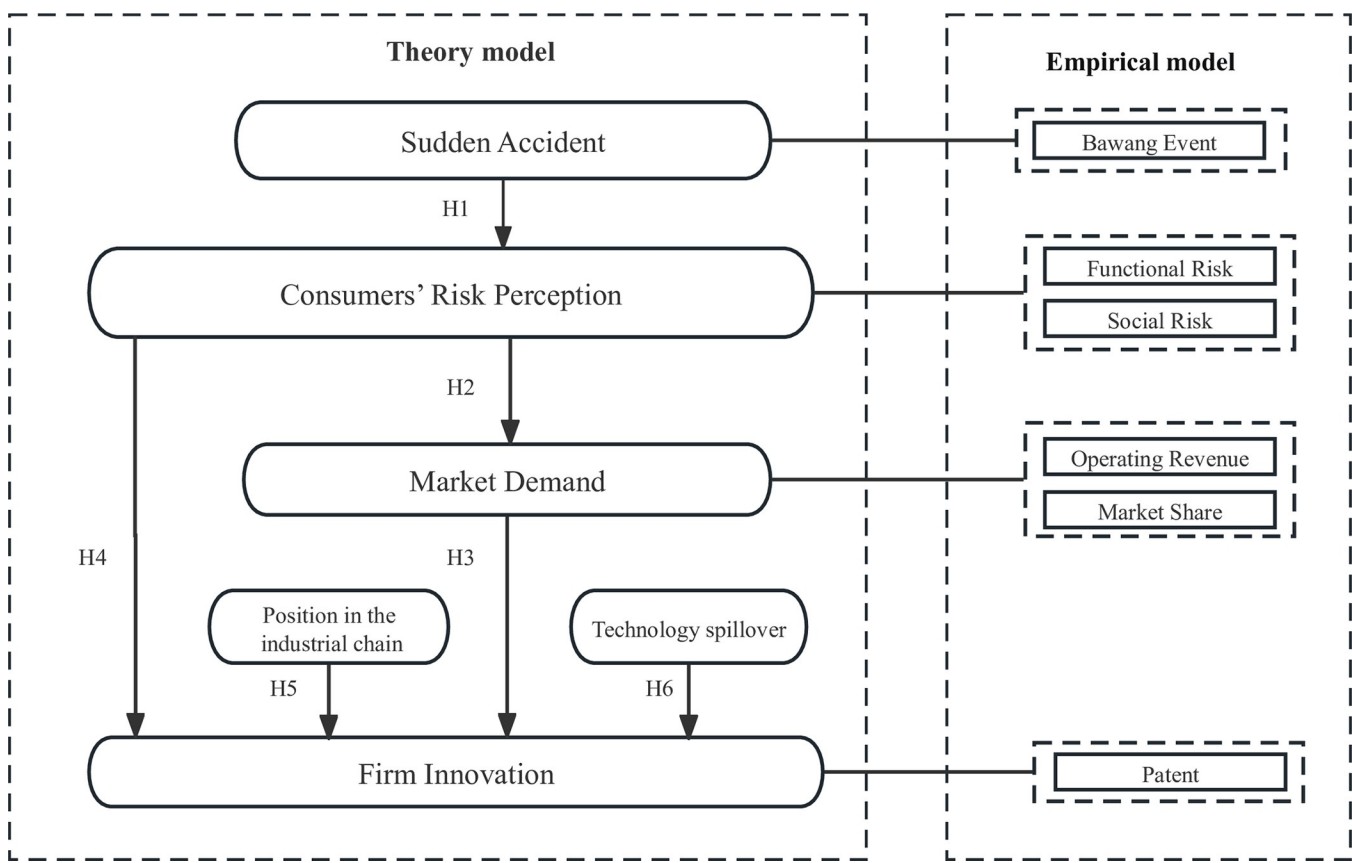

**Fig 1. Framework of the research.**

Based on the preceding discussion, this research is based on the idea that when a sudden accident occurs, such as a product harm crisis, consumers' risk perception tends to increase, affecting their purchasing behavior. This change in consumer behavior leads to a shift in market demand, eventually influencing manufacturing enterprises' innovation activities. We establish our theoretical and empirical analysis frameworks, illustrated in Fig 1.

## 3. Event background

### 3.1. Bawang Shampoo Causes Cancer news

On July 14, 2010, Next Magazine published an article titled "Bawang Shampoo Causes Cancer". The article reported that the Hong Kong General Public Inspection Office had conducted tests which revealed that Bawang shampoo contained dioxane at 27 parts per million (ppm). Dioxane is classified as a carcinogen by the United States, and its content in Bawang shampoo exceeds the maximum standard of 10 ppm set by California. The article explained that dioxane is dangerous because it can enter the bloodstream through the skin and cause liver and kidney damage, skin eruptions, immune system damage, and even death if inhaled excessively. The report caused widespread concern and was widely reported by mainstream media and websites. This led to a heated debate among consumers and the media. Following the report, the Consumer Council of Hong Kong tested 60 shampoos for dioxane and found that more than 60% contained dioxane. Seven of the shampoos exceeded the safety standard recommended by

the Scientific Committee on Consumer Safety (SCCS) of the European Union. This further fueled the debate among consumers and the media [48].

The Bawang Shampoo Causes Cancer Event, referred to as the "Bawang event," had severe financial consequences for the Bawang Group. In 2017, Bawang's operating revenue was 107 million yuan, a 20% decrease from the previous year, and their primary product, Bawang Shampoo, had an operating revenue of 87.59 million yuan, a 17.2% decrease from the previous year. The Bawang Group's market value plummeted to H.K. $800 million, a 96% reduction from its peak value, and Bawang's leading position in the market vanished. On December 27, 2017, Bawang suspended its stock trading on the market [49].

## 3.2. Changes in consumers' perceived risk

Consumers face uncertainty when making a purchase decision, and this uncertainty is known as risk [11]. Many studies show that perceived risk is an essential factor that affects consumer behavior [50,51]. The perceived risk associated with a product can reduce consumer trust and discourage its purchase and use [52–54]. Perceived risk can be divided into different categories, such as financial, functional, physical, psychological, social, and time risk [26,55,56]. R The specific dimension selected for research may vary depending on the focus of the study [57,58]. In this section, we explore consumers' risk perception changes from two perspectives, using the Bawang event as an example.

**3.2.1. Social risk.**  Baidu is a leading player in China's online search industry, with a significant market share and many users. The Baidu Search Index measures public interest in specific keywords based on Baidu web and news searches. It uses search volume data to analyze the weight of each keyword's frequency in Baidu web searches. Past research has shown that the Baidu Search Index can be used to understand consumer behavior. The Baidu Search Index reflects public opinion and social risks related to consumers [59,60]. This study uses the Baidu Search Index to examine shifts in how consumers perceive social risk. Before the Bawang event on July 14, 2010, the daily average Baidu Search Index values for "Bawang" and "Dioxane" were 823 and 95, respectively. However, after the incident, the daily average Baidu Search Index values skyrocketed to 6,621 and 3,752, respectively. The Baidu index's surge indicates that consumers suddenly became more aware of the carcinogenic risks of Bawang shampoo and other dioxane-related products, which raised their perceived social risk of using these products.

**3.2.2. Functional risk.**  When a product fails to meet consumers' expectations or performs worse than its competitors, it increases the functional risk dimension in consumers' perception of risk. This often results in consumers buying less of the product. For example, a survey conducted on Sina about Bawang shampoo's carcinogens showed that 73.6% of 35,367 netizens who voted said they would not continue to buy Bawang shampoo. In comparison, only 12.6% said they would. Sohu health news section reporters in Changsha, Carrefour, Metro, and other supermarkets randomly interviewed many consumers, and most of them said they would refuse to buy or suspend their use of Bawang shampoo products. Similarly, a reporter from Guangzhou Daily learned that some stores that initially sold Bawang shampoo did not remove the products from their shelves but all carried out sales promotions [61]. This indicates that the Bawang brand's image has been damaged, and many consumers' willingness to buy its products has been weakened. Additionally, the overlord incident has significantly increased consumers' perceived risk.

## 4. Data and identification strategy

### 4.1. Data

The data used in this study to measure firm innovation mainly come from the China Firm Innovation Integration Database and the Innography Database. These databases provide

information on the innovation activities of Chinese firms during the sample period, including the number of patent applications, registered trademarks, research fund expenditure, and IPC classification. Patents are commonly used in research to measure firm innovation due to their apparent advantages [62]. For instance, patents are the innovation output, reflecting firms' input and resource utilization efficiency and innovation ability. Additionally, patent data contains comprehensive information such as the applicant, technology category, and legal status, which can be utilized to study firm innovation behavior in detail by matching other firm-level data [63]. Therefore, this study uses the number of invention patent applications to measure firms' innovation.

The firm-level data used in this study is derived from the Annual Survey of Industrial Firms (ASIF), which includes all state-owned and private firms with annual sales exceeding 5 million yuan. The variables chosen as control variables at the firm level were selected based on ensuring the homogeneity of the control variables as much as possible while considering the availability of relevant data. To maintain consistency with previous studies, this research uses the firm scale (measured by the natural logarithm of total assets), asset-liability ratio (measured by the ratio of total liabilities to total assets), return on assets (measured by the ratio of net profit to total assets), the scale of fixed assets (measured by the ratio of fixed assets to total assets), and the age of the firm (calculated by subtracting the number of years since the firm was established from the current year).

This study utilized the Chinese Firm Innovation Integration Database, Innography Database, and ASIF to obtain data on innovation and other information about Chinese firms from the period of 2008 to 2014. The choice of this particular interval was based on two reasons. Firstly, the Bawang event occurred in 2010, and using data from 2008 to 2014 would meet the requirements of the method on samples and ensure accurate and effective estimation results. Secondly, China implemented the new drug approval reform in 2015, which significantly impacted China's pharmaceutical innovation. Therefore, data from 2015 and later are unsuitable for inclusion in our research sample.

The descriptive statistics of the main variables are reported in Table 1.

## 4.2. Identification strategy

This paper employs a difference-in-differences approach by using the Bawang event as a quasi-natural experiment to identify the impact of consumers' risk perception changes on firm innovation.

The accidental nature of the Bawang event and the extensive media coverage at the time provide quite convincing evidence of the externality of our shock. As we mentioned in Chapter

**Table 1. Summary statistics.**

| Variable | Observation | Mean | Std. Dev. | Min | Max |
| --- | --- | --- | --- | --- | --- |
| lnpatent | 239,262 | 0.731 | 1.058 | 0 | 10.487 |
| Size | 237,384 | 11.466 | 1.641 | 0 | 20.672 |
| Lev | 237,234 | 0.695 | 0.896 | 0.031 | 7.937 |
| ROA | 237,153 | 0.129 | 0.346 | -109.482 | 29.085 |
| TFA | 236,744 | 0.374 | 0.729 | 0 | 14.328 |
| Age | 309,923 | 11.758 | 9.169 | 0 | 65 |

Note: This table reports the summary statistics of the variables used in this paper. lnpatent is the natural logarithm of the number of invention patent applications plus 1. Size is total assets, Lev is the book value of total debt divided by the market value of total assets, ROA is operating income divided by total assets, TFA is total fixed assets, and Age is the number of years a firm has been in existence since its founding.

3 of this article, after the Bawang event on July 14, 2010, media coverage of Bawang shampoo and dioxane spiked. Furthermore, the Baidu search trend for "Bawang shampoo" and "dioxane" suggests that public interest in these topics increased dramatically after the Bawang event.

During the Bawang event, media coverage drew attention to carcinogenic Dioxane in Bawang shampoo, which increased consumers' risk perception. Dioxane is a commonly used chemical in the production of medicine, daily chemicals, and other unique fine chemical products. Therefore, not only shampoos but also other products like toothpaste, deodorant, mouthwash, cosmetics, and pharmaceuticals may contain dioxane. In addition, it is worth mentioning that the daily chemical industry is closely connected to the pharmaceutical manufacturing industry. Both industries require similar raw materials and technologies for their products. Many pharmaceutical companies also manufacture daily chemical products. We have listed some Chinese pharmaceutical companies that do so in S1 Table. Due to the externality of the perceived risk, the impact of the Bawang event was not limited to the Bawang alone but also extended to the entire daily chemical and pharmaceutical industry. We have chosen the pharmaceutical and daily chemical industries as the experimental group, while the other manufacturing industries will serve as the control group. Based on the industry classification code, we have selected firms from the following industries as samples for the processing group, as shown in S2 Table. The empirical specification is set as follows:

$$lnpatent_{ipt} = \beta_0 + \beta_1 Treat_i \times After2010_t + \alpha X_{it-1} + \delta_i + \delta_p + \delta_t + \varepsilon_{ipt} \tag{1}$$

where $i$ represents the firm, $p$ represents the province, and $t$ represents the time. $lnpatent_{ipt}$ is the dependent variable of this model, which is measured by the natural logarithm of the number of invention patent applications plus 1. $Treat_i$ is a dummy variable that takes the value of 1 when the firm belongs to the treatment group and 0 otherwise. $After2010_t$ is also a dummy variable that takes the value of 1 after 2010 (the year of Bawang event) and 0 otherwise. $\beta_1$ is the research interest of this research. Specifically, if $\beta_1 > 0$ and is statistically significant, it indicates that the increase in consumers' risk perception has a positive effect on firms' innovation. By contrast, if $\beta_1 < 0$ and is statistically significant, it indicates that the increase in consumers' risk perception has a negative effect on firms' innovation. $X_{it-1}$ are a vector of control variables at the firm level, including firm scale, asset-liability ratio, return on assets, fixed assets scale, and firm age. All control variables are lagged by one year to address the potential endogeneity issues. $\delta_i$ is the firm fixed effects, $\delta_p$ is the province fixed effects, and $\delta_t$ is the year fixed effects. $\varepsilon_{ipt}$ is the stochastic error term. In order to address the issue take care of autocorrelation, standard errors are clustered at the firm level.

## 5. Empirical analysis

### 5.1. Baseline regressions

Table 2 presents the baseline regression results that show the impact of changes in consumers' risk perception on innovation. Column (1) represents the baseline specification, including all variables except the year and province fixed effects. In column (2), the year fixed effects are added, and the results are similar to column(1) but with a smaller coefficient. In column (3), the province fixed effects are added, leading to the preferred specification regarding the goodness-of-fit, represented by the R-squared value. The regression coefficient of the interaction is statistically significant and positive, with a magnitude of 0.076. This indicates that after the Bawang event, compared to the control group, the number of patent applications in the treatment group increased by 7.6%. Overall, the baseline regression results indicate that an increase in consumers' risk perception incentivizes firms' innovation. A sudden accident leads to decreased consumer demand, causing the firm's profits and market share to suffer. In such

Table 2. Baseline regression results.

| Dependent variable | lnpatent | lnpatent | lnpatent |
|---|---|---|---|
| Treat × After2010 | 0.241*** | 0.053*** | 0.076*** |
| | (0.005) | (0.021) | (0.022) |
| Size | 0.161*** | 0.054*** | 0.057*** |
| | (0.005) | (0.006) | (0.006) |
| Lev | 0.053*** | 0.020*** | 0.021*** |
| | (0.004) | (0.004) | (0.004) |
| ROA | 0.022* | -0.020* | -0.018 |
| | (0.012) | (0.012) | (0.012) |
| TFA | 0.007*** | 0.007* | 0.007* |
| | (0.001) | (0.005) | (0.005) |
| Age | 0.017*** | -0.001 | -0.001 |
| | (0.001) | (0.001) | (0.001) |
| Firm FE | Yes | Yes | Yes |
| Year FE | No | Yes | Yes |
| Province FE | No | No | Yes |
| R-squared | 0.031 | 0.073 | 0.074 |
| Observation | 146,821 | 146,821 | 146,821 |

Note: Standard errors reported in parentheses are clustered at the firm level.

*, **, and *** indicate significance at the 10%, 5%, and 1% levels, respectively.

situations, firms need to take measures to regain consumer trust and reduce their perception of risk. However, changing factor input or withdrawing from the market can be costly for firms. Therefore, implementing technological or product innovation is their best option [64]. The positive impact of such innovation can also lead to the innovation of related firms and contribute to the overall growth of the industry's innovation ability.

## 5.2. Event study

The DID approach requires fulfilling the parallel trend assumption before it can be used. The treatment and control groups should exhibit a similar trend in the absence of any policy shock. This means that the two groups should be comparable. To test this assumption, an event study framework was implemented in this study. Based on Eq (1), this study includes year dummy variables, $Yrdum_m$, and their interactions with the dummy variable $Treat_i$, as:

$$lnpatent_{ipt} = \beta_0 + \sum\nolimits_{m=2008}^{2013} \beta_1 Treat_i \times Yrdum_m + \alpha X_{it-1} + \delta_i + \delta_p + \delta_t + \varepsilon_{ipt} \qquad (2)$$

Fig 2 presents the dynamic effects of the Bawang event on firm innovation using the event study design. The estimated coefficients were statistically insignificant before the Bawang event, which satisfied the parallel trend assumption. After the Bawang event, the estimated coefficients become significant, and their magnitudes gradually increase. In summary, the test shows that the difference between the treatment group and the control group did not change over time before the Bawang event and that there is no evidence to suggest that firms in the treatment group own more patents.

## 5.3. Placebo test

This study has randomly selected both the treatment and control groups to ensure that non-random selection does not influence the results. The regression analysis has been repeated

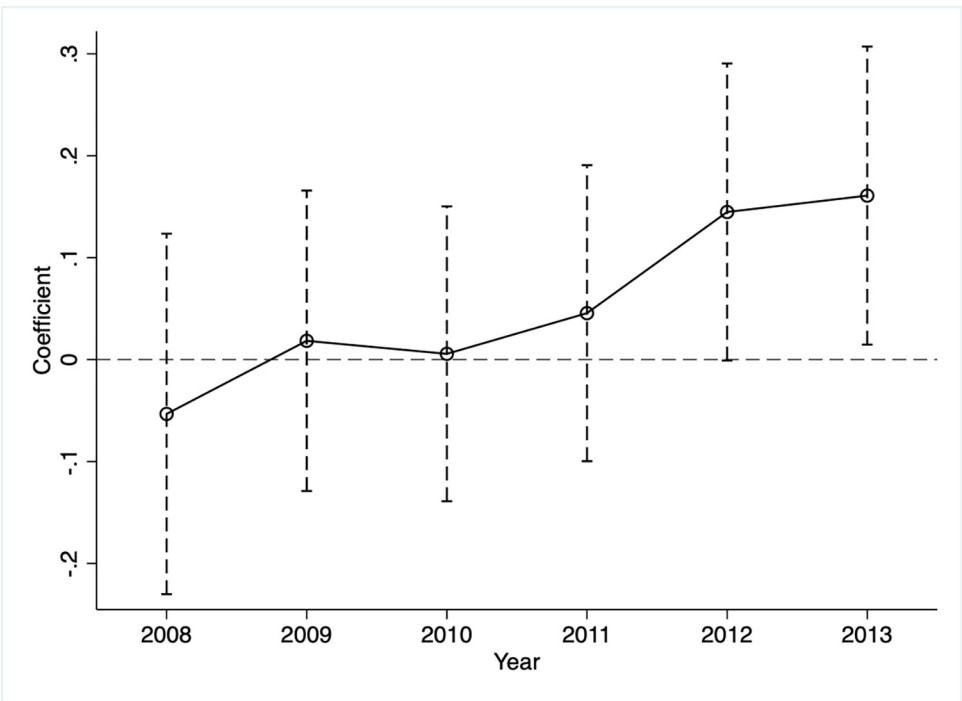

**Fig 2. Event study of the effect of the risk perception on firm innovation.** Note:The hollow circles in the Fig 2 are the estimated coefficients, and the vertical bars are the 90% confidence intervals of the estimated coefficients.

1,000 times using this random sample, and the estimated distribution of the $\hat{\beta}_{random}$ is shown in Fig 3. The estimated coefficient $\hat{\beta}_{random}$ has an insignificant mean of -0.004, which is very close to 0. This contrasts the real estimate value of 0.076, far from 0. These results suggest that the baseline regression results are unlikely to be driven by chance.

### 5.4. Robustness test

**5.4.1. PSM-DID.** To overcome bias in the Difference-in-Differences (DID) estimation and address systematic variations in innovation behavior between control and treatment groups, we utilized the Propensity Score Matching (PSM)-DID method. We obtained propensity scores by carrying out logit regression of covariates using the dummy variable $Treat_i$. The firms whose propensity score was closest to the treatment group were paired to form the control group. To ensure the effectiveness of the PSM-DID method, the study carried out diagnosis tests. Logit regression results revealed that firm size, asset-liability ratio, return on assets, firm size of fixed assets, and firm age have strong explanatory power for the indicator of treatment. The study also conducted tests to determine whether the distribution of each variable becomes balanced between the treatment and control groups after matching. Table 3 shows no systematic difference between the treatment group and the control group after matching. The study also drew the propensity score value density function graph to test the matching effect between the treatment and control groups. Fig 4 indicates that the probability densities of the propensity score values of the treatment group are very close to the control group, demonstrating the validity of the matching process. Column (1) of Table 4 shows the regression results using the PSM-DID method, which aligns with the baseline regression results, thus further supporting the conclusion of this paper that consumers' risk perception has a significant and positive effect on firm innovation.

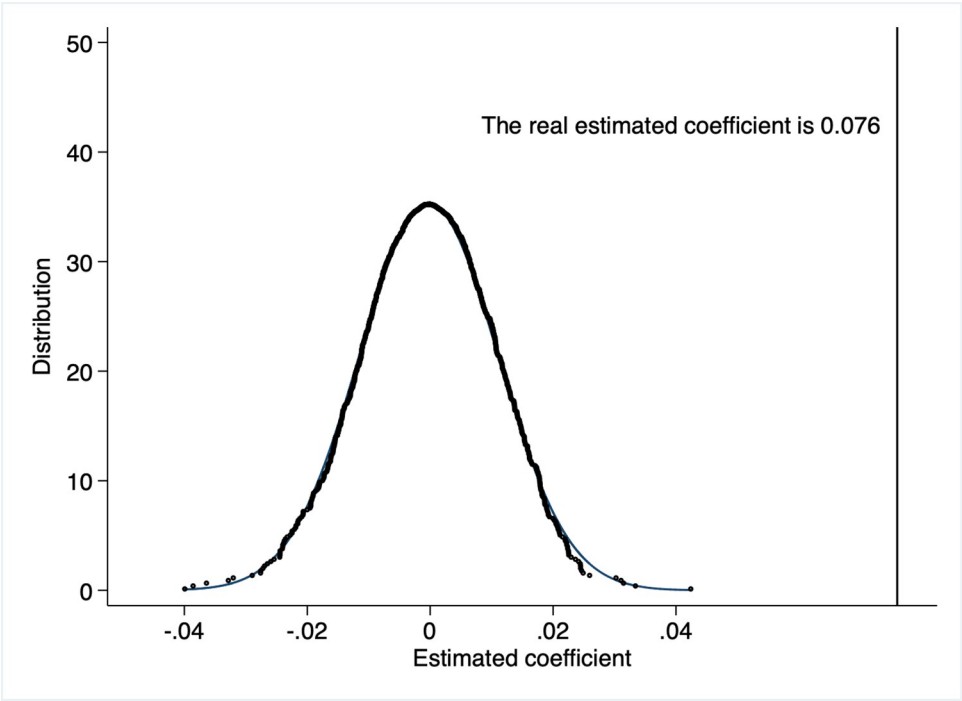

**Fig 3. Placebo test.**

**5.4.2. Alternative dependent variable.** This study uses the number of invention patents as the dependent variable in the baseline regression. However, due to the strict approval criteria for patents, this variable only partially represents the actual innovation situation. The number of registered trademarks representing innovation output is used to complementarily measure innovation, following the previous literature [65,66]. Therefore, this study takes the information on the trademark as the dependent variable. The results in column (2) of Table 4 indicate that the increase in consumers' risk perception promotes firms' trademarks, which aligns with the baseline results.

## 5.5. Heterogeneous tests

**5.5.1. Ownership.** In order to ensure the robustness of the regression results, this paper investigates the impact of consumers' risk perception on the innovation activities of firms with different ownership structures. We divided the total sample into two sub-samples, state-owned firms, and private firms, according to the type of firm registration, and estimated Eq (1) separately for each. The results are shown in Table 5. The regression coefficient of the interaction

**Table 3. The PSM validity test.**

| Variable | Treatment group mean | Control group mean | Difference | *t*-value |
|---|---|---|---|---|
| Size | 11.674 | 11.719 | -0.045 | -0.78 |
| Lev | 0.472 | 0.475 | -0.003 | -0.30 |
| ROA | 0.124 | 0.120 | 0.004 | 0.68 |
| TFA | 0.330 | 0.330 | -0.000 | -0.04 |
| Age | 13.137 | 13.087 | 0.050 | 0.12 |

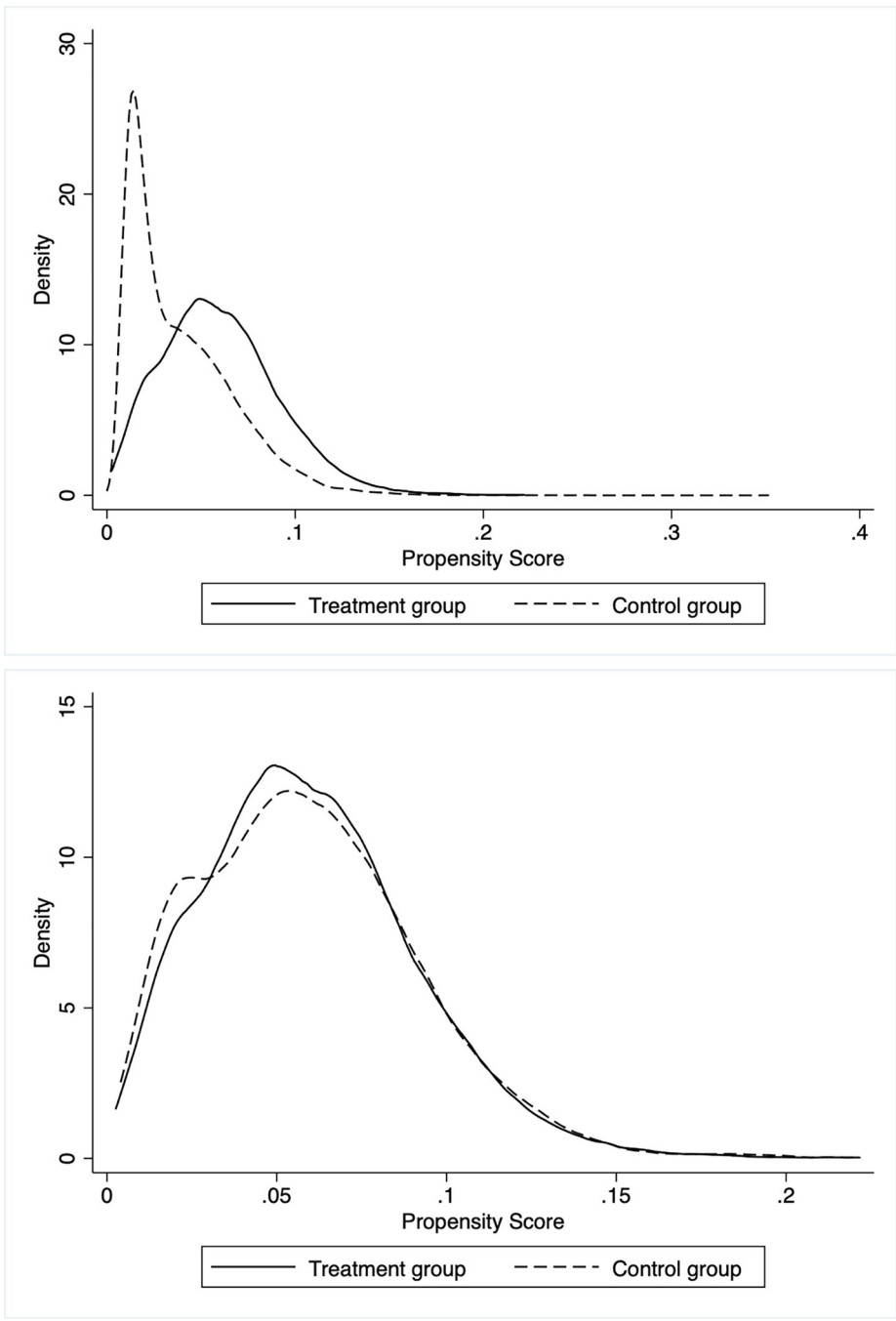

**Fig 4.** Nuclear density of matching; (a) Propensity score before matching; (b) Propensity score after matching.

term in column (2) is statistically significant and positive, while insignificant in column (1). It indicates that the increase in consumers' risk perception has a more significant impact on the innovation of private firms than state-owned firms.

State-owned firms are usually financially strong, with resource advantages and government support. They have a stable market position, which makes it easier for them to maintain stability during a crisis. As a result, they are less interested in high-risk and long-cycle R&D projects.

**Table 4. Robustness test.**

| Dependent variable | lnpatent | lntrademark |
|---|---|---|
| Treat × After2010 | 0.076*** | 0.117*** |
| | (0.022) | (0.027) |
| Control variables | Yes | Yes |
| Firm FE | Yes | Yes |
| Year FE | Yes | Yes |
| Province FE | Yes | Yes |
| R-squared | 0.075 | 0.024 |
| Observation | 68,998 | 145,696 |

Note: Control variables include Size, Lev, ROA, TFA, Age. Standard errors reported in parentheses are clustered at the firm level.

*, **, and *** indicate significance at the 10%, 5%, and 1% levels, respectively.

On the other hand, private firms have flexible decision-making mechanisms and efficient resource allocation. They prioritize innovation in the face of market competition and crises, quickly responding to market demand and promoting technological progress and innovative development [67].

**5.5.2. Geographic location.** In order to ensure the robustness of the regression results, this paper investigates the impact of consumers' risk perception on the innovation activities of firms with different geographic locations. The China Statistical Yearbook categorizes China's 34 provinces into three regions, namely, eastern, central, and western, based on location. We divided our research sample into three sub-samples according to the location of the firm province and conducted the regression analysis. The results are presented in Table 6. The regression coefficient of the interaction term in column (1) is statistically significant and positive, indicating that the increase in consumers' risk perception positively impacts the innovation of firms in the eastern region. However, the coefficients in columns (2) and (3) are negative and insignificant, suggesting that the increase in consumers' risk perception does not significantly affect firms' innovation in the western and central regions.

The eastern region of China is more developed compared to other regions. Its residents have better living conditions, the cost of obtaining information is lower, and consumers give

**Table 5. Heterogeneous effects of ownership.**

| Subsample | State-owned | Private |
|---|---|---|
| Dependent variable | lnpatent | lnpatent |
| Treat × After2010 | 0.077 | 0.078*** |
| | (0.113) | (0.023) |
| Control variables | Yes | Yes |
| Firm FE | Yes | Yes |
| Year FE | Yes | Yes |
| Province FE | Yes | Yes |
| R-squared | 0.173 | 0.072 |
| Observation | 5,032 | 141,631 |

Note: Control variables include Size, Lev, ROA, TFA, Age. Standard errors reported in parentheses are clustered at the firm level.

*, **, and *** indicate significance at the 10%, 5%, and 1% levels, respectively.

**Table 6. Heterogeneous effects of location.**

| Subsample | Eastern | Central | Western |
|---|---|---|---|
| Dependent variable | lnpatent | lnpatent | lnpatent |
| Treat × After2010 | 0.074*** | -0.013 | -0.012 |
| | (0.024) | (0.050) | (0.082) |
| Control variables | Yes | Yes | Yes |
| Firm FE | Yes | Yes | Yes |
| Year FE | Yes | Yes | Yes |
| Province FE | Yes | Yes | Yes |
| R-squared | 0.066 | 0.118 | 0.164 |
| Observation | 119,226 | 19,411 | 8,254 |

Note: Control variables include Size, Lev, ROA, TFA, Age. Standard errors reported in parentheses are clustered at the firm level.

*, **, and *** indicate significance at the 10%, 5%, and 1% levels, respectively.

more importance to the safety of products. In case of a product harm crisis, consumers' risk perception in the eastern region tends to be higher than those in the central and western regions. Consequently, firms operating in the eastern region experience more significant fluctuations in market demand, which often leads to a higher willingness to innovate [68].

## 6. Mechanism

The above analysis shows that the Bawang event led to a significant increase in innovation activities. This section provides direct evidence for our proposed mechanism—i.e., consumers' risk perception impacts market demand. Our empirical strategy relies on a standard difference-in-differences estimation:

$$Demand_{ipt} = \beta_0 + \beta_1 Treat_i \times After2010_t + \alpha X_{it-1} + \delta_i + \delta_p + \delta_t + \varepsilon_{ipt} \qquad (3)$$

where $i$ represents the firm, $p$ represents the province, and $t$ represents the time. $Demand_{ipt}$ is the dependent variable of this model, which is measured by the firm's sales (main business income) and market share (proportion of a firm's sales in the whole industry). $Treat_i$ is a dummy variable that takes the value of 1 when the firm belongs to the treatment group and 0 otherwise. $After2010_t$ is also a dummy variable that takes the value of 1 after 2010 (the year of the Bawang event) and 0 otherwise. $X_{it-1}$ is a vector of control variables at the firm level, including firm scale, asset-liability ratio, return on assets, fixed assets scale, and firm age. $\delta_i$ is the firm fixed effects, $\delta_p$ is the province fixed effects, and $\delta_t$ is the year fixed effects. $\varepsilon_{ipt}$ is the stochastic error term.

The results are shown in Table 7. The regression coefficient of the interaction term in columns (1) and (2) are statistically significant. This means that when the Bawang even occurs, the increase in consumer perceived risk will lead to a decline in market demand for related products. Companies will conduct research and innovation activities to address consumers' doubts about existing technologies and products. This will accelerate product upgrades and ensure the production of safer and higher-quality products.

After the product injury crisis, numerous negative coverage emerged regarding the companies and products involved, which could lead to severe physical and mental harm to consumers and cause feelings of tension, fear, and other negative emotions. At this time, consumers' risk perception of related products will increase significantly. Their willingness to buy and use such products will decrease significantly, and the demand for safer and higher quality products will increase, ultimately leading to a change of demand in the entire market [28–31]. Given

**Table 7. Risk perception and market share.**

| Dependent variable | Sales income | Market share |
|---|---|---|
| Treat × After2010 | -1.506*** | -0.012*** |
| | (0.264) | (0.004) |
| Control variables | Yes | Yes |
| Firm FE | Yes | Yes |
| Year FE | Yes | Yes |
| Province FE | Yes | Yes |
| R-squared | 0.017 | 0.032 |
| Observation | 236,506 | 140,955 |

Note: Control variables include Size, Lev, ROA, TFA, Age. Standard errors reported in parentheses are clustered at the firm level.

*, **, and *** indicate significance at the 10%, 5%, and 1% levels, respectively.

that a large amount of literature proves that demand can affect innovation activities [69,70], our theoretical hypothesis that risk perception affects innovation by influencing market demand is valid.

# 7. Further discussion

In this section, we introduce economic concepts such as industrial chain, industrial correlation, and technology spillover effect into the research of "demand-driven innovation" within the framework of risk perception.

## 7.1. The position in the industrial chain

The industrial chain is the connection between firms in different industries. The upstream industry is located at the top of the industrial chain and provides necessary raw materials and primary products to the downstream industry. The downstream industry is mainly involved in processing and modifying raw materials to convert them into actual consumer products. As downstream firms are closer to the market and consumers, they are more sensitive to the changes in consumer demand. Therefore, they are more responsive to changes in risk perception during product crises and carry out innovative activities more actively [71,72].

After considering the discussion above, we propose Hypothesis 5, which suggests that changes in risk perception impact firm innovation, and the extent of the impact depends on the firm's position in the industrial chain. To test Hypothesis 5, this study calculates the level of upstreamness of 153 industries by using the input-output table of China according to the method of Antràs et al. (2012) [73].

$$Upstreamness_i = 1 \times \frac{F_i}{Y_i} + 2 \times \frac{\sum_{j=1}^N \hat{\mu}_{ij} F_j}{Y_i} + 3 \times \frac{\sum_{j=1}^N \sum_{k=1}^N \hat{\mu}_{ik} \hat{\mu}_{kj} F_j}{Y_i} + 4 \times \frac{\sum_{j=1}^N \sum_{k=1}^N \sum_{l=1}^N \hat{\mu}_{il} \hat{\mu}_{lk} \hat{\mu}_{kj} F_j}{Y_i} + \cdots \quad (4)$$

where, $\hat{\mu}_{ij}$ represents the output of industry i required to produce 1 unit value of industry j's output (the marker above indicates adjustments for open economies and inventories), $F_i$ represents the portion of the output of industry j that is used for final consumption, and $Y_i$ represents the total output of industry i. Each item on the right side of the equation corresponds to a production link in the industrial chain with varying distances from the final consumption. The numbers to the left of the multiplication sign, i.e., 1,2,3, etc., represent the distance from final consumption plus one. The part to the left of the multiplication sign in each term represents the portion of the output of industry i that is used in the corresponding position in the industrial chain, as the weight exists. The sum of the items gives the $Upstreamness_i$, it represents the weighted average distance between the output of industry i and final consumption. A higher degree of

upstream indicates that the industry is closer to intermediate inputs along the industrial chain, and vice versa, closer to the final product.

Using the index $Upstreamness_i$ to measure the industrial position of a firm, this study proposes a triple differences model to discuss the above question, as:

$$lnpatent_{ipt} = \beta_0 + \beta_1 Treat_i \times After2010_t \times Upstreamness_i + \beta_2 Treat_i \times After2010_t$$
$$+ \beta_3 After2010_t \times Upstreamness_i + \alpha X_{it-1} + \delta_i + \delta_p + \delta_t + \varepsilon_{ipt} \quad (5)$$

The results are shown in column (1) of Table 8. The estimated coefficient is significant and negative, which means that the change of consumers' risk perception has a positive impact on firm innovation, and the extent of the impact depends on the firm's position in the industrial chain. Furthermore, the lower the $Upstreamness_i$, the greater the motivation for firms to innovate. In other words, downstream firms are more involved in innovative activities.

The government needs to pay attention to the impact of consumers' risk perception on different positions within the manufacturing industry chain. The government should adopt differentiated policies, focusing on stimulating innovation for industries with weak innovation capacity through tax incentives, subsidies, and other policies. At the same time, the government should improve the merger and reorganization mechanisms of firms, eliminate firms with low technical levels, and optimize the industry through resource integration and factor remaking. For firms with strong innovation ability, the government should guide and encourage firms to transform external factors, such as consumers' risk perception, into internal driving forces for their development model through policy preferences.

## 7.2. Spillover

Technological advancements in one industry can positively impact the productivity of other industries through spillover effects. When adding R&D input into the production function, it is essential to consider not only the R&D input of the industry itself but also the interindustry technology spillover effect caused by using products as carriers [74,75]. Industries with high technological similarity can learn from each other's new technologies and innovative ideas to promote industry innovation. Thus, cross-industry technology spillover is more likely to occur between firms and industries with high similarity [76]. In the case of the Bawang event, the daily chemical industry was the most affected. Changes in consumers' risk perception will impact companies with technologies similar to those used in the daily chemical industry. In light of this discussion, we propose Hypothesis 6, which suggests that changes in risk perception impact firm innovation, which is related to technology spillover and technical similarity. To test Hypothesis 6, according to Hohberger et al. (2015) [77], this study calculated $Dist_i$, the Euclidean distance between the firm's patent portfolio and the industry's patent portfolio as a whole. The formula used for this calculation is as follows:

$$Dist_i = \sqrt{\sum \left( P_{ki} - P_k \right)^2}$$

where $Dist_i$ represents the Euclidean distance between the firm and industry patent portfolio, $P_{ki}$ refers to the proportion of firm i's patents in class k in all patents, and $P_k$ refers to the industry where firm i is located, and the proportion of patents in class k in all patents in this industry.

**Table 8. Industrial position and spillovers.**

| Dependent variable | lnpatent | lnpatent |
|---|---|---|
| Treat × After2010 × Upstreamness | -0.479*** (0.143) | |
| Treat × After2010 × Dist | | -0.225** (0.108) |
| Treat × After2010 | 1.250*** (0.357) | 0.081*** (0.023) |
| Treat × Upstreamness | 0.827*** (0.242) | |
| Treat × Dist | | 0.214 (0.250) |
| After2010 × Upstreamness | -0.004 (0.004) | |
| After2010 × Dist | | 0.216*** (0.055) |
| Control variables | Yes | Yes |
| Firm FE | Yes | Yes |
| Year FE | Yes | Yes |
| Province FE | Yes | Yes |
| R-squared | 0.074 | 0.075 |
| Observation | 146,821 | 147,726 |

Note: Control variables include Size, Lev, ROA, TFA, Age. Standard errors reported in parentheses are clustered at the firm level.

*, **, and *** indicate significance at the 10%, 5%, and 1% levels, respectively.

We add an index $Dist_i$ to measure the technical similarity and estimate the following model,

$$lnpatent_{ipt} = \beta_0 + \beta_1 Treat_i \times After2010_t \times Dist_i + \beta_2 Treat_i \times After2010_t$$
$$+ \beta_3 Treat_i \times Dist_i + \beta_4 After2010_t \times Dist_i + \alpha X_{it-1} + \delta_i + \delta_p + \delta_t + \varepsilon_{ipt} \quad (6)$$

As shown in column (2) of Table 8, the estimated coefficient of triple interaction is statistically significant and negative. This indicates that during a product crisis, due to the technology spillover effect, the firm whose technology is closer to the relevant industry or firm will be more affected by consumers' perceived risks. This will also make them more willing to carry out innovative activities. This study also calculates the similarity between industrial sectors according to the method of Los (2000) and Frenken et al. (2007) [78,79]; the calculation steps are reported in S1 Appendix. This study obtains consistent empirical results using the alternative variable.

The government should utilize industrial policy to encourage collaboration among firms, universities, and scientific research institutions. This will help exchange advanced technologies and innovative ideas and expand the technology spillover effect between firms. This will rapidly improve China's manufacturing enterprises' overall independent research and development and technological innovation capabilities.

## 8. Conclusions and recommendations

### 8.1. Main conclusion

In this paper, we examine the links between consumers' risk perception, market demand, and firm innovation, taking advantage of the disclosure and the extensive reporting of the Bawang event in 2010. The theoretical hypothesis section of this paper argues that when a sudden

accident occurs, such as a product harm crisis, consumers' risk perception tends to increase, which, in turn, affects their purchasing behavior. This change in consumer behavior leads to a shift in market demand, eventually influencing manufacturing enterprises' innovation activities. This paper employs a difference-in-differences approach by using the Bawang event as a quasi-natural experiment to identify the impact of consumers' risk perception changes on firm innovation. The findings are as follows.

First, consumers' risk perception effectively promotes firms to innovate. The increase in consumers' risk perception of products containing dioxane after the Bawang event led to a significant increase in patent applications in the daily chemical and pharmaceutical manufacturing sectors, which rose by approximately 7.6% compared to other manufacturing industries. We propose that consumers' risk perception affects firm innovation by influencing market demand and provide direct evidence to support this mechanism. In a sudden accident, consumer demand can decrease, leading to a firm's profits and market share decline. Firms need to take appropriate measures to regain consumer trust and reduce their perception of risk. However, changing factor input or withdrawing from the market can be expensive for firms. Thus, implementing technological or product innovation is their best choice [64].

Second, we examine the impact of the position in the industrial chain and technology spillover on consumers' perceived risk and firm innovation. Our study has revealed two findings: First, within the same industry chain, the downstream industry is more vulnerable to changes in consumer demand due to its closer proximity to consumers and, therefore, more sensitive to risk perception. This sensitivity makes changes in innovation activities more noticeable in the downstream industry. Second, cross-industry technology spillovers are more likely to occur between firms and industries with high technical similarity when a sudden accident occurs. The higher the technological similarity with the firm involved, the more significant the impact of consumers' risk perception on the innovation of the enterprise.

## 8.2. Possible marginal contribution

Compared with the existing research, the possible marginal contribution of this paper is as follows.

1. The first is the innovation of the research perspective. Our research focuses on the relationship between market demand, innovation, and consumers' risk perception. While there is ample literature on how market demand drives innovation, there is little research on how consumers' risk perception affects demand-driven innovation. Our research aims to fill this gap and provide a comprehensive understanding of the subject. In addition, our research also examines the strategies that firms can adopt to restore consumer trust and reduce risk perception following product harm crises. While previous studies have mainly focused on external strategies, such as crisis PR and advertising, internal strategies, such as improving product quality and technology through innovation, can also be crucial in restoring consumer trust and reducing perceived risks. Given the unstable macroeconomic environment in China, it is essential to examine how external shocks affect firm innovation and R&D. Our study has practical significance and policy implications for firms operating in such conditions.

2. The second is the innovation of research methods. Previous studies on "demand-driven innovation" have not been successful in solving the endogenous problem and obtaining an unbiased and consistent estimate. However, this paper addresses this issue by selecting the Bawang event as an exogenous shock and using the DID approach method to avoid endogeneity problems.

3. The third is the innovation of theoretical mechanisms. This paper introduces economic concepts such as industrial chain, industrial correlation, and technology spillover effect in the research framework on the impact of consumers' risk perception on firm innovation. Our research is the first study that finds a relationship between the influence of consumers' risk perception on firm innovation and their position in the industrial chain and technical similarity.

## 8.3. Recommendations

This paper's research content and empirical results have policy implications for the real world today, mainly in three ways.

1. For the government. This paper examines how consumers' risk perception affects firm innovation and how the industrial chain location and technological similarity influence it. The study suggests that when the government formulates industrial and innovation policies, it needs to consider the role of the industrial chain and the spillover effect. The government should create matching policy programs for different types of firms. On the one hand, the government should prioritize the development of leading and high-impact industries and increase investment and policy support for them. This approach would promote the joint development of other sectors. On the other hand, the government should recognize that some firms may have weak innovation ability and enthusiasm and struggle to cope with demand-side shocks. To encourage them, the government can stimulate the internal motivation of these firms to innovate through tax incentives, subsidies, and other policies. Additionally, the government should improve the merger and reorganization mechanism of firms. It should eliminate firms with low technical levels and optimize industries through resource integration and factor reconstruction.

2. For the firm. Firms should take advantage of the spillover effect of innovation and participate actively in collaborative innovation activities within their supply chain while optimizing their innovation strategies. Specifically, firms should closely monitor the innovation dynamics of related companies and learn from and absorb their innovation results to improve their product quality and production efficiency, thereby enhancing their competitiveness. Additionally, firms should maintain timely and effective interaction with industry-related firms to keep track of their innovation plans and the development progress of new products. By collaborating with other firms and sharing resources and knowledge, firms can improve their innovation ability and performance. For instance, they can jointly establish joint research and development teams with other firms to develop new technologies and products, reducing costs, shortening research and development cycles, and improving market competitiveness.

3. For the consumers. This paper's research results indicate that consumers' perceived risk can influence market demand, affecting firms' innovation. Therefore, both governments and firms need to pay heed to the consumers. Amidst the intensifying downward risks of the global economy, the government should focus on promoting household consumption and expanding domestic demand. To achieve this, the government can stimulate consumers' willingness to consume by providing them with cash subsidies, consumer vouchers, and other incentives. When the market demand is strong, there are sufficient incentives for firms to invest in research and development projects, improve production efficiency, and increase product diversity, thus promoting economic growth. Firms should listen to consumers' opinions and feedback during production, operation, and research and

development and amend their business strategy and research and development direction accordingly. On the other hand, consumers should actively exercise and maintain their right to supervision, right to know, and other relevant rights and collaborate with the government to supervise and give feedback on relevant firms and products. This would encourage firms to provide better products and services to the consumers.

## 8.4. Research deficiencies and prospects

This study comprehensively and systematically discusses the relationship between consumers' perceived risks, market demand, and firm innovation. However, there is room for improvement in this study due to some limitations, such as the availability of data, subjective academic research ability, and understanding level.

The paper primarily concentrates on the Bawang event and two specific industries, namely the pharmaceutical industry and the daily chemical industry. The empirical results obtained have a limited application scope. Nevertheless, the research ideas and theoretical framework can be applied to similar issues, such as the impact of changes in consumers' perceived risks on market demand and innovation when other unexpected events occur. Take COVID-19, for example. The outbreak of COVID-19 in early 2020 has had a significant impact on countries worldwide due to its rapid and widespread transmission and the difficulties in preventing and controlling it. Consumer demand has changed after the outbreak of COVID-19 due to increased risk perception. Residents have increased their spending on medical supplies to prevent the epidemic, leading to increased demand for products and services in the medical and healthcare fields. Additionally, residents have minimized going out and opted to isolate at home to prevent the spread of the virus, thus shifting their demand from offline to online platforms such as online retail, fresh food e-commerce, express logistics, online entertainment, online education, and remote work. The changes in consumers' risk perception and behavior have forced relevant industries to carry out technological innovations to cope with changes in people's needs after the epidemic. The COVID-19 pandemic has impacted various industries and regions. In the future, researchers can utilize it as an external shock to analyze the effect of shifts in consumers' perceived risks on market demand and innovation during unexpected events. These findings will have broader applicability once data becomes available.

This paper analyzes explicitly the changes in risk perception from the perspectives of social risk and functional risk, and it is more detailed than previous studies. However, like previous studies, this paper's discussion on consumers' risk perception is still a qualitative analysis. In the future, constructing quantitative indicators of consumers' risk perception into the empirical model will become a focus of subsequent research.

## Supporting information

**S1 Table. Category of the treatment group.**
(DOCX)

**S2 Table. Daily chemical products produced by pharmaceutical firms.**
(DOCX)

**S1 Appendix. Calculating similarity between industrial sectors.**
(DOCX)

**S1 Data.**
(ZIP)

## Acknowledgments

The authors thank Dr. Lihan Jiang, Prof. Shiyuan Pan, Prof. Bing Ye, Prof. Zibin Zhang, Prof. Jianlian Ye and Prof. Limin Du for their invaluable help and advice throughout the study. We appreciate the two anonymous reviewers and our academic editors for their thoughtful criticisms and suggestions.

## Author Contributions

**Conceptualization:** Jing Cao, Haiwei Jiang, Xiaomeng Ren, Jinchuan Shi.

**Data curation:** Jing Cao.

**Investigation:** Jing Cao.

**Methodology:** Jing Cao, Haiwei Jiang, Xiaomeng Ren.

**Software:** Jing Cao.

**Supervision:** Jinchuan Shi.

**Writing – original draft:** Jing Cao.

**Writing – review & editing:** Jing Cao, Haiwei Jiang, Xiaomeng Ren.

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
