## [Decision Letter · Decision Letter 0]

16 Jan 2024

PONE-D-23-31451Consumers’ Risk Perception, Market Demand, and Firm Innovation: Evidence from ChinaPLOS ONE

Dear Dr. Ren,

Thank you for submitting your manuscript to PLOS ONE. After careful consideration, we feel that it has merit but does not fully meet PLOS ONE’s publication criteria as it currently stands. Therefore, we invite you to submit a revised version of the manuscript that addresses the points raised during the review process.

We look forward to receiving your revised manuscript.

Kind regards,

Abaid Ullah Zafar, Ph.D.

Academic Editor

PLOS ONE

2. For studies involving third-party data, we encourage authors to share any data specific to their analyses that they can legally distribute. PLOS recognizes, however, that authors may be using third-party data they do not have the rights to share. When third-party data cannot be publicly shared, authors must provide all information necessary for interested researchers to apply to gain access to the data. (https://journals.plos.org/plosone/s/data-availability#loc-acceptable-data-access-restrictions)

a) A description of the data set and the third-party source

b) If applicable, verification of permission to use the data set

c) Confirmation of whether the authors received any special privileges in accessing the data that other researchers would not have

d) All necessary contact information others would need to apply to gain access to the data

4. Please ensure that you include a title page within your main document. You should list all authors and all affiliations as per our author instructions and clearly indicate the corresponding author.

Reviewers' comments:

Reviewer's Responses to Questions

**Comments to the Author**

1. Is the manuscript technically sound, and do the data support the conclusions?

Reviewer #1: Yes

Reviewer #2: Partly

2. Has the statistical analysis been performed appropriately and rigorously? 

Reviewer #1: Yes

Reviewer #2: Yes

3. Have the authors made all data underlying the findings in their manuscript fully available?

Reviewer #1: Yes

Reviewer #2: Yes

4. Is the manuscript presented in an intelligible fashion and written in standard English?

Reviewer #1: Yes

Reviewer #2: No

5. Review Comments to the Author

Reviewer #1: This research article explores the relationship between consumers' risk perception and firm innovation, using the Bawang Shampoo Causes Cancer Event as a case study. While the study provides some valuable insights, there are several aspects to consider such as:

1)The paper inconsistently uses different citation styles (e.g., some references are in APA style, while others follow an alternative style). A consistent and proper citation style should be used throughout the paper.

2)The conclusion provides a brief summary of the research, but it could be more focused on summarizing the key findings and their implications. It would also be beneficial to highlight limitations and suggest directions for future research

3)The article briefly mentions policy implications but could provide a more detailed discussion of how the findings could inform business strategies, government policies, and consumer behavior. Practical recommendations based on the research findings would enhance the article's relevance.

4) The paper proposes a theoretical framework but does not deeply explain the theoretical underpinnings of the study, especially in the context of consumer behavior and innovation. It would be helpful to discuss how consumer risk perception aligns with established theories in economics and consumer psychology. The choice of the "difference-in-differences" (DID) approach is suitable

5)There are minor language issues, such as grammatical errors and awkward sentence structures, that could be addressed for improved clarity and readability

6)The paper focuses on a specific event, the Bawang Shampoo Causes Cancer Event in China. This limited scope might hinder the generalizability of the findings to other contexts. The paper should discuss the limitations and potential implications for the applicability of the results to other regions or industries

Reviewer #2: Thank you for the opportunity to review this paper. The authors adequately provide the details and explain each section of the manuscript well. Overall, the design and method of this study is sound. However, the authors need to address a couple of issues before the manuscript reaches the publishable level.

1. The introduction is not clear and focused. The concepts presentation needs more connection to improve the text flow. Furthermore, the reference to the research gap could be clearer. I suggest starting with a brief overall topic characterization, secondly, giving some detail to the main concepts, and finally, focusing on the research gap and research question (or research objectives). The argumentation for the selected topic must be based on recent literature.

2. The current literature review is weak. It should be further extended. Literature review did not reveal any literature gap this study attempted to address, even though author(s) have mentioned the literature gap in the implication section.

3. Literature review covers several topics relevant for the research scope. However, a better connection between the topics is required. Additionally, the author(s) must further develop each topic as the text emerges in a fragmented way. Therefore, the argumentation for the hypotheses' formulation is weak and I suggest that it could be revised. I suggest that more specific relations could be inferred from the literature.

4. The research methodology needs to be cleaned up. More details on methodology should be provided.

5. The discussion on the research findings is relatively simple and insufficient. I recommend to elaborate the findings focusing on the study context and compare the research results with prior studies in order to manifest the contribution of this study. The discussion section needs significantly more depth. It should include subsections (e.g., theoretical implications, managerial implications, limitations and suggestions for future research). The generalizability of the results should be better discussed.

6. The paper is required for proper writing and needs to be revised with professional academic writing. There are several grammatical issues throughout the paper.

6. PLOS authors have the option to publish the peer review history of their article (what does this mean?). If published, this will include your full peer review and any attached files.

Reviewer #1: No

Reviewer #2: No

---

## [Author Response · Author response to Decision Letter 0]

4 Mar 2024

Response to Reviewer #1: 

This research article explores the relationship between consumers' risk perception and firm innovation, using the Bawang Shampoo Causes Cancer Event as a case study. While the study provides some valuable insights, there are several aspects to consider such as:

Reply: We thank the anonymous reviewer very much for reading the manuscript so carefully, and offering so many important and constructive comments. We have benefited tremendously from your constructive comments and suggestions, which are all extremely helpful to bring the paper up to the standard. We have tried to make changes and revisions to make the research theme more prominent and the logic clearer. Please allow us to take this opportunity to send our greatest appreciation to you. We try to address these problems and the flaws you point out in the following.

Question 1: The paper inconsistently uses different citation styles (e.g., some references are in APA style, while others follow an alternative style). A consistent and proper citation style should be used throughout the paper.

Reply: We sincerely thank the reviewer for careful reading. We have meticulously revised the citation style and ensured consistent citation style throughout the manuscript. We believe these changes contribute to the overall clarity and professionalism of the document.

Question 2: The conclusion provides a brief summary of the research, but it could be more focused on summarizing the key findings and their implications. It would also be beneficial to highlight limitations and suggest directions for future research.

 Reply: Thanks for this valuable comment. We have revised this part according to your suggestion (pp.19-22). 

(i) We have now summarized our research content and methods, and explained our main empirical findings and their implications in the conclusion part.

“ In this paper, we examine the links between consumers’ risk perception, market demand, and firm innovation, taking advantage of the disclosure and the extensive reporting of the Bawang event in 2010. The theoretical hypothesis section of this paper argues that when a sudden accident occurs, such as a product harm crisis, consumers’ risk perception tends to increase, which, in turn, affects their purchasing behavior. This change in consumer behavior leads to a shift in market demand, eventually influencing manufacturing enterprises’ innovation activities. This paper employs a difference-in-differences approach by using the Bawang event as a quasi-natural experiment to identify the impact of consumers’ risk perception changes on firm innovation. The findings are as follows.

First, consumers’ risk perception effectively promotes firms to innovate. The increase in consumers’ risk perception of products containing dioxane after the Bawang event led to a significant increase in patent applications in the daily chemical and pharmaceutical manufacturing sectors, which rose by approximately 7.6% compared to other manufacturing industries. We propose that consumers’ risk perception affects firm innovation by influencing market demand and provide direct evidence to support this mechanism. In a sudden accident, consumer demand can decrease, leading to a firm’s profits and market share decline. Firms need to take appropriate measures to regain consumer trust and reduce their perception of risk. However, changing factor input or withdrawing from the market can be expensive for firms. Thus, implementing technological or product innovation is their best choice. 

Second, we examine the impact of the position in the industrial chain and technology spillover on consumers’ perceived risk and firm innovation. Our study has revealed two findings: First, within the same industry chain, the downstream industry is more vulnerable to changes in consumer demand due to its closer proximity to consumers and, therefore, more sensitive to risk perception. This sensitivity makes changes in innovation activities more noticeable in the downstream industry. Second, cross-industry technology spillovers are more likely to occur between firms and industries with high technical similarity when a sudden accident occurs. The higher the technological similarity with the firm involved, the more significant the impact of consumers’ risk perception on the innovation of the enterprise.”

(ii) We have discussed the contributions and limitations of our research, and future research directions (pp.20-22).

Compared with the existing research, we have provided the possible marginal contribution of our manuscript (1) The first is the research perspective. While there is ample literature on how market demand drives innovation, there is little research on how consumers’ risk perception affects demand-driven innovation. Our research aims to fill this gap and provide a comprehensive understanding of the subject. In addition, our research also examines the strategies that firms can adopt to restore consumer trust and reduce risk perception following product harm crises. While previous studies have mainly focused on external strategies, such as crisis PR and advertising, we find that internal strategies, such as improving product quality and technology through innovation, can also play a crucial role in restoring consumer trust and reducing perceived risks. (2) The second is the research methods. Previous studies on "demand-driven innovation" have not been successful in solving the endogenous problem and obtaining an unbiased and consistent estimate. However, this paper addresses this issue by selecting the Bawang event as an exogenous shock and using the DID approach method to avoid endogeneity problems. (3) The third is the theoretical mechanisms. This paper introduces economic concepts such as industrial chain, industrial correlation, and technology spillover effect in the research framework on the impact of consumers’ risk perception on firm innovation. It is the first study that finds a relationship between the influence of consumers’ risk perception on firm innovation and their position in the industrial chain and technical similarity.

We have also discussed the limitations of our research and the future research directions. During our discussion, we explored the extent to which the results we obtained can be applied or generalized to other contexts. While our empirical findings may have limited applicability, the research concepts and theoretical framework can still be used to investigate similar issues. For instance, we can apply them to study the impact of changes in consumers' perceived risks on market demand and innovation during unexpected events like the COVID-19 pandemic (pp.21-22). In addition, this paper specifically analyzes the changes in risk perception from the perspectives of social risk and functional risk, and it is more detailed than previous studies. However, like previous studies, the discussion on consumers’ risk perception in this paper is still a qualitative analysis. In the future, constructing quantitative indicators of consumers’ risk perception into the empirical model will become a focus of subsequent research.

Question 3: The article briefly mentions policy implications but could provide a more detailed discussion of how the findings could inform business strategies, government policies, and consumer behavior. Practical recommendations based on the research findings would enhance the article's relevance.

 Reply: Thank you for this suggestion. We have revised this section to elaborate on the policy implications based on our findings (pp.20-21). 

“The research content and empirical results of this paper have policy implications for the real world today, mainly in three ways.

(1) For the government. This paper examines how consumer risk perception affects firm innovation and how the industrial chain location and technological similarity influence it. The study suggests that when the government formulates industrial and innovation policies, it needs to consider the role of the industrial chain and the spillover effect. The government should create matching policy programs for different types of firms. On the one hand, the government should prioritize the development of leading and high-impact industries and increase investment and policy support for them. This approach would promote the common development of other sectors. On the other hand, the government should recognize that some firms may have weak innovation ability and enthusiasm and may struggle to cope with demand-side shocks. To encourage them, the government can stimulate the internal motivation of these firms to innovate through tax incentives, subsidies, and other policies. Additionally, the government should improve the merger and reorganization mechanism of firms. It should eliminate firms with low technical levels and optimize industries through resource integration and factor reconstruction.

 (2) For the firm. Firms should take advantage of the spillover effect of innovation and participate actively in collaborative innovation activities within their supply chain while optimizing their innovation strategies. Specifically, firms should closely monitor the innovation dynamics of related companies and learn from and absorb their innovation results to improve their product quality and production efficiency, thereby enhancing their competitiveness. Additionally, firms should maintain timely and effective interaction with industry-related businesses to keep track of their innovation plans and the development progress of new products. By collaborating with other firms and sharing resources and knowledge, firms can improve their innovation ability and performance. For instance, they can jointly establish joint research and development teams with other firms to develop new technologies and products, reducing costs, shortening research and development cycles, and improving market competitiveness. 

(3) For the consumers. This paper's research results indicate that consumers' perceived risk can influence market demand, affecting firms' innovation. Therefore, both governments and firms need to pay heed to the consumers. Amidst the intensifying downward risks of the global economy, the government should focus on promoting household consumption and expanding domestic demand. To achieve this, the government can stimulate the willingness of consumers to consume by providing them with cash subsidies, consumer vouchers, and other incentives. When the market demand is strong, there are sufficient incentives for firms to invest in research and development projects, improve production efficiency, and increase product diversity, thus promoting economic growth. Firms should listen to consumers' opinions and feedback during production, operation, and research and development and amend their business strategy and research and development direction accordingly. On the other hand, consumers should actively exercise and maintain their right to supervision, right to know, and other relevant rights and collaborate with the government to supervise and give feedback on relevant firms and products. This would encourage firms to provide better products and services to the consumers.”

Question 4: The paper proposes a theoretical framework but does not deeply explain the theoretical underpinnings of the study, especially in the context of consumer behavior and innovation. It would be helpful to discuss how consumer risk perception aligns with established theories in economics and consumer psychology. The choice of the "difference-in-differences" (DID) approach is suitable.

 Reply: Thanks for this helpful comment. We have revised Section 2 (pp.3-7) of our research paper according to the reviewer's suggestion. 

Our theoretical framework is based on the idea that when a sudden accident occurs, such as a product harm crisis, consumers’ risk perception tends to increase, which, in turn, affects their purchasing behaviour. This change in consumer behaviour leads to a shift in market demand, eventually influencing manufacturing enterprises’ innovation activities. Our theoretical underpinnings are developed based on this logic.

First, we have reviewed relevant literature on consumer psychology to explore how sudden accidents impact consumers' perceived risk, affect their behavior, and ultimately influence market demand. Based on our findings, we have proposed a research hypothesis:

Hypothesis 1 (H1). Sudden accident has an impact on consumers’ risk perception. 

Hypothesis 2 (H2). Consumers’ risk perception has an impact on market demand.

Then, we have delved into the relevant economics literature to explore how demand influences innovation. We will then present our research hypothesis:

Hypothesis 3 (H3). Market demand has an impact on firm innovation.

Finally, we have examined how consumers' risk perception drives innovation as a demand-pulling force, highlighting its differences from other demand-pulling forces. Furthermore, we have introduced economic concepts such as industrial chain, industrial linkage, and technology spillover into the research of "demand-driven innovation" under the framework of risk perception.

Hypothesis 4 (H4). Consumers’ risk perception has an impact on firm innovation.

Hypothesis 5 (H5). Consumers’ risk perception has an impact on firm innovation, and the extent of the impact depends on the firm’s position in the industrial chain.

Hypothesis 6 (H6). Consumers’ risk perception has an impact on firm innovation and the extent of the impact related to technology spillover.

Through the discussion of these three topics, we have connected consumers’ risk perception with the literature on consumer psychology and economics. Our research has covered several topics, and to establish a better connection between them, we have divided Section 2 into three parts: "Consumers' Risk Perception and Market Demand", "Market Demand and Firm Innovation", and "Consumers' Risk Perception and Firm Innovation". In each section, we have explained our theoretical underpinnings and relevant concepts, reviewed and commented on research about relevant topics, and presented our research hypotheses and theoretical frameworks. We have not presented the full contents of the revisions here. However, they can be found in the revised paper (pp.3-7). 

We chose the DID approach as our research method for the following reasons. Previous studies on "demand-driven innovation" have not been successful in solving the endogenous problem and obtaining an unbiased and consistent estimate. Our paper addresses this issue by selecting the Bawang event as an exogenous shock and using the DID approach method to avoid endogeneity problems. The accidental nature of the Bawang event and the extensive media coverage at the time provide quite convincing evidence of the externality of our shock.

Question 5: There are minor language issues, such as grammatical errors and awkward sentence structures, that could be addressed for improved clarity and readability.

Reply: We apologize for the language issue of our manuscript. We tried our best to improve the manuscript and made some changes to the manuscript. We did not list the changes here but marked them in red in the revised paper. We appreciate for reviewer's warm work earnestly and hope that the correction will meet with approval.

Question 6: The paper focuses on a specific event, the Bawang Shampoo Causes Cancer Event in China. This limited scope might hinder the generalizability of the findings to other contexts. The paper should discuss the limitations and potential implications for the applicability of the results to other regions or industries.

Reply: Thank you for this suggestion. We have added a paragraph in the conclusion to discuss our research's limitations and the possible directions for future research (pp.21-22).

“The paper primarily concentrates on the Bawang event and two specific industries, namely the pharmaceutical industry and the daily chemical industry. The empirical results obtained have a limited application scope. Nevertheless, the research ideas and theoretical framework can be applied to similar issues, such as the impact of changes in consumers' perceived risks on market demand and innovation when other unexpected events occur. Take COVID-19, for example. The outbreak of COVID-19 in early 2020 has had a significant impact on countries worldwide due to its rapid and widespread transmission and the difficulties in preventing and controlling it. Consumer demand has changed after the outbreak of COVID-19 due to increased risk perception. Residents have increased their spending on medical supplies to prevent the epidemic, leading to an increase in demand for products and services in the medical and healthcare fields. Additionally, residents have minimized going out and opted to isolate at home to prevent the spread of the virus, thus shifting their demand from offline to online platforms such as online retail, fresh food e-commerce, express logistics, online entertainment, online education, and remote work. The changes in consumers' risk perception and behaviour have forced relevant industries to carry out technological innovations to cope with changes in people's needs after the epidemic. The COVID-19 pandemic has impacted various industries and regions. In the future, researchers can utilize it as an external shock to analyze the effect of shifts in consumers' perceived risks on market demand and innovation during unexpected events. These findings will have broader applicability once data becomes available.”

Finally, we hope that the changes we have made solve all your concerns about the article. We are more than happy to make any further changes that will improve the paper. Thank you again for your time and help.

Response to Reviewer #2: 

Thank you for the opportunity to review this paper. The authors adequately provide the details and explain each section of the manuscript well. Overall, the design and method of this study is sound. However, the authors need to address a couple of issues before the manuscript reaches the publishable level.

Reply: We thank the anonymous reviewer very much for reading the manuscript so carefully, and offering so many important and constructive comments. We have benefited tremendously from your constructive comments and suggestions, which are all extremely helpful to bring the paper up to the standard. We have tried to make changes and revisions to make the research theme more prominent and the logic clearer. Please allow us to take this opportunity to send our greatest appreciation to you. We try to address these problems and the flaws you point out in the following.

Question 1: The introduction is not clear and focused. The concepts presentation needs more connection to improve the text flow. Furthermore, the reference to the research gap could be clearer. I suggest starting with a brief overall topic characterization, secondly, giving some detail to the main concepts, and finally, focusing on the research gap and research question (or research objectives). The argumentation for the selected topic must be based on recent literature.

Reply: Thanks for this helpful comment. We have revised the introduction part of our research paper according to the reviewer's suggestion (pp.2-3).

The following is the train of thought behind writing the introduction of our paper. Firstly, we introduced the research background of this paper, leading to our research theme, "Consumers' Risk Perception, Market Demand, and Firm Innovation". We have discussed the relevant literature on demand-induced innovation and proposed that although scholars have examined the mechanism of demand-induced innovation, only a few have explored the role of consumers' risk perception as a demand-side influencing factor. Our research aims to fill this gap.

We then explained the concept and characteristics of consumers' risk perception, highlighting how it differs from other demand-inducing forces. “Consumers’ risk perception refers to the uncertainty that a consumer may feel when making a purchase decision. They may not be sure whether the purchase results will meet their requirements, and sometimes, the results may not be pleasant. When a sudden accident occurs, such as a product harm crisis, negative information about the product and the firm involved spreads rapidly in the market. This causes widespread concern among consumers, increases their risk perception and ultimately affects their purchase decisions. This change in consumer behaviour leads to a shift in market demand, which in turn affects the innovation activities of manufacturing enterprises. Changes in consumers’ risk perception can have a significant impact on firm innovation, and this impact can differ from other demand-induced forces. There are several reasons why this is so. Firstly, it is often difficult for consumers to have access to all the information they need about risks, and their understanding of product risks can be influenced by small probability events and media overexposure. Secondly, the impact of increased risk perception on firm innovation is ambiguous. When consumers perceive more risk, their willingness to pay for security increases and the demand for safer products also goes up. This, in turn, may lead to an increase in the willingness of firm to innovate products and technologies. However, on the other hand, the willingness to innovate on products and technologies that are considered high-risk may decline. Thirdly, changes in risk perception also have an externality on firm innovation. Negative events can impact not only the companies directly involved but also the businesses and industries involved. Given these specificities of consumers’ risk perception, research on the effect of consumers’ risk perception on firm innovation can be a valuable addition to the existing literature on "demand-induced innovation".”

Next, we briefly introduced our main research contents, methods, and data and explained why we chose them. “This paper uses the relevant data of Chinese firms from 2008 to 2014, taking Chinese consumers, Chinese manufacturing firms, and the Chinese market as research objects, employs a difference-in-differences(DID) approach by using the Bawang event, a sudden accident, as a quasi-natural experiment to identify the impact of changes in consumers’ risk perception on firm innovation. The accidental nature of the Bawang event and the extensive media coverage at the time provide quite convincing evidence of the externality of our shock. We have decided to utilize the data from 2008 to 2014 for two main reasons. Firstly, the Bawang event occurred in 2010, which makes it reasonable to use data from 2008 to 2014. This ensures the accuracy and effectiveness of estimation results in accordance with the method on samples. Secondly, there was a new drug approval reform in China in 2015 which significantly impacted China's pharmaceutical innovation. Therefore, data from 2015 onwards is not suitable for inclusion in our research sample.”

To demonstrate its contribution, we also compared our research with the latest literature on perceived risk and innovation, namely Galasso and Luo (2019). “There are some differences between our research and that of Galasso and Luo(2021). Firstly, compared with Galasso and Luo, we have provided a more detailed discussion on changes in consumers’ perceived risk. We have a comprehensive and detailed discussion on two aspects of consumers’ perceived risk: social risk and functional risk. Secondly, this paper introduces economic concepts such as industrial chain, industrial correlation, and technology spillover effect into the research of “demand-driven innovation” within the framework of risk perception. Our research indicates that the influence of consumers’ risk perception on firm innovation depends on their position in the industrial chain and technology spillover. We have two main findings: (1) Firms closer to the downstream of the industrial chain are more sensitive to changes in consumers’ perceived risks, leading to a greater impact on their innovation activities. (2) Cross-industry technology spillovers are more likely to occur between firms and industries with high technical similarity when a sudden accident occurs. The higher the technological similarity with the firm involved, the greater the impact of consumers’ risk perception on the innovation of the firm. Thirdly, the study conducted by Galasso and Luo analyzes the course of technological progress using patent and FDA data at the patent classification level. Meanwhile, we analyze Chinese manufacturing firms as micro subjects of innovation activities. We focus on the decisions made when responding to negative events to recover their reputation and compensate for the losses incurred. We also add a discussion on firm heterogeneity to enhance the analysis.”

Finally, we presented the structure of our paper. For details, please see the revisions (pp.2-3).

Question 2: The current literature review is weak. It should be further extended. Literature review did not reveal any literature gap this study attempted to address, even though author(s) have mentioned the literature gap in the implication section.

 Reply: Thanks for this helpful comment. We have revised Section 2 (pp.3-7) of our research paper according to the reviewer's suggestion. Our literature is divided into three parts: "Consumers' Risk Perception and Market Demand", "Market Demand and Firm Innovation", and "Consumers' Risk Perception and Firm Innovation". We have conducted a detailed review of the literature on the relevant topic for each section. After each literature review, we identified the literature gaps we will address in our work.

 The literature gaps that exist in the theme of "Consumers' Risk Perception and Market Demand" are: “Although some scholars have studied how changes in consumers’ risk perception and trust affect their purchasing decisions following a sudden accident, such as a product harm crisis, there is still a need for more comprehensive discussions on the strategies that firms should adopt to restore consumers’ trust and reduce their risk perception. Previous studies mainly focused on external strategies, such as crisis public relations and advertising, and overlooked internal strategies, such as improving products and technology through innovation.”

 The literature gaps that exist in the theme of "Market Demand and Firm Innovation" are: “This paper believes that there are still some problems worthy of further discussion. First, we can discuss how market demand drives innovation from a richer perspective. For example, changes in market demand caused by risk perception have received little empirical and theoretical attention. Second, the endogeneity problem has yet to be well solved in previous studies, and it is not easy to get an unbiased and consistent estimate. This paper holds that the most effective way to study the relationship between market demand and firm innovation is to find a reasonable exogenous impact and estimate it using the quasi-natural experiment method. Third, industrial heterogeneity and industry correlation have been neglected in previous studies. The same exogenous impact will have different effects on the innovation behaviour of different industries. Even in the same industrial chain, the industry in the upstream and the industry in the downstream of the industrial chain will have different changes in innovation behaviour when the market demand changes. In addition, due to the inter-industry correlation and technology spillover effect, the change of innovation behaviour in one industry will affect the innovation of other industries, which is also worth noting.”

 The literature gaps that exist in the theme of "Consumers' Risk Perception and Firm Innovation" are: “The literature most directly related to our study is Galasso and Luo (2021). While we have borrowed their ideas and methods, there are some ways in which our research differs from that of Galasso and Luo. Firstly, Galasso and Luo noted that over-radiation accidents and their extensive media coverage can lead to an increased perception of risk regarding medical radiation among patients and medical providers. However, they did not provide a detailed analysis of how the perception of risk changes over time. We have chosen to discuss two aspects of consumers’ perceived risk, namely social risk and functional risk, and provide a more detailed explanation of the changes in consumers’ perceived risk over time. Secondly, this paper introduces economic concepts such as industrial chain, industrial correlation, and technology spillover effect into the research of "demand-driven innovation" within the framework of risk perception. This has yet to be done in previous studies. We have presented our two hypotheses and conducted experiments to verify them: (1) Consumers’ risk perception impacts firm innovation, and the extent of the impact depends on the firm’s position in the industrial chain. (2) Consumers’ risk perception impacts firm innovation and the extent of the impact related to technology spillover. Thirdly, the study conducted by Galasso and Luo analyzes the course of technological progress using patent and FDA data at the patent classification level. Meanwhile, we analyze Chinese manufacturing firms as micro subjects of innovation activities, focusing on the decisions made when responding to negative events to recover their reputation and make up for the losses incurred. We also add a discussion on firm heterogeneity to enhance the analysis.”

Question 3: Literature review covers several topics relevant for the research scope. However, a better connection between the topics is required. Additionally, the author(s) must further develop each topic as the text emerges in a fragmented way. Therefore, the argumentation for the hypotheses' formulation is weak and I suggest that it could be revised. I suggest that more specific relations could be inferred from the literature.

Reply: Thank you for this suggestion. We have revised Section 2 (pp.3-7) of our research paper according to the reviewer's suggestion. Our research has covered several topics, and to establish a better connection between them, we have divided Section 2 into three parts: "Consumers' Risk Perception and Market Demand", "Market Demand and Firm Innovation", and "Consumers' Risk Perception and Firm Innovation". In each section, we have explained our theoretical underpinnings and relevant concepts, reviewed and commented on research about relevant topics, and presented our research hypotheses and theoretical frameworks.

In the section on "Consumers' Risk Perception and Market Demand", we have discussed in detail the concept of consumers' risk perception, how sudden accidents can affect consumer behaviour and how consumer behaviour can affect market demand. In the section on "Market Demand and Firm Innovation", we have explained how market demand can influence firm innovation. In the section on "Consumers' Risk Perception and Firm Innovation", we have explained how consumers' risk perception can influence firm innovation, and we have highlighted the differences between consumers' risk perception and other demand-driven forces that impact firm innovation. We have also explained how our research on perceived risk and innovation differs from the latest literature by Galasso and Luo (2019). Through the discussion of these three topics, we have connected consumer risk perception with the literature on consumer psychology and economics.

We have not presented the full contents of the revisions here. However, they can be found in the revised paper (pp.3-7).

Question 4: The research methodology needs to be cleaned up. More details on methodology should be provided.

Reply: Thank you for this suggestion. We have added more details to the methodology in our manuscript, specifically in the following areas:

 In section 4.2, we have explained the identification strategy we use in more detail (pp.10-11). “The accidental nature of the Bawang event and the extensive media coverage at the time provide quite convincing evidence of the externality of our shock. After the Bawang event on July 14, 2010, media coverage of Bawang shampoo and dioxane spiked. Furthermore, the Baidu search trend for "Bawang shampoo" and "dioxane" suggests that public interest in these topics increased dramatically after the Bawang event. During the Bawang event, media coverage drew attention to the presence of carcinogenic Dioxane in Bawang shampoo, which increased consumers’ risk perception. Dioxane is a commonly used chemical in the production of medicine, daily chemicals, and other special fine chemical products. Therefore, not only shampoos but also other products like toothpaste, deodorant, mouthwash, cosmetics, and pharmaceuticals may contain dioxane. Due to the externality of the perceived risk, the impact of the Bawang event was not limited to the Bawang alone but also extended to the entire daily chemical and pharmaceutical industry. We have chosen the pharmaceutical and daily chemical industries as the experimental group, while the other manufacturing industries will serve as the control group. Based on the industry classification code, we have selected firms from the following industries as samples for the processing group, as shown in Table A.2.”.

 In Section 6, we have supplemented the details of the methods and empirical models used (pp.16). “Our empirical strategy relies on a standard difference-in-differences estimation: 

〖〖Demand〗_ipt=β〗_0+β_1 〖Treat〗_i×〖After2010〗_t+αX_(it-1)+δ_i+δ_p+δ_t+ε_ipt (3) 

where i represents the firm, p represents the province, and t represents the time. 〖Demand〗_ipt is the dependent variable of this model, which is measured by the firm’s sales (main business income) and market share (proportion of a firm’s sales in the whole industry).〖Treat〗_j is a dummy variable that takes the value of 1 when the firm belongs to the treatment group and 0 otherwise. 〖After2010〗_t is also a dummy variable that takes the value of 1 after 2010 (the year of the Bawang event) and 0 otherwise. X_(it-1) is a vector of control variables at the firm level, including firm scale, asset-liability ratio, return on assets, fixed assets scale, and firm age. δ_i is the firm fixed effects, δ_p is the province fixed effects, and δ_t is the year fixed effects. ε_ipt is the stochastic error term.”.

 In Section 7, we have detailed the method of measuring the variable 〖Upstreamness〗_i (pp.17-18). “To test Hypothesis 5, this study calculates the level of upstreamness of 153 industries by using the input-output table of China according to the method of Antràs et al. (2012).

〖Upstreamness〗_i=1×F_i/Y_i +2×(∑_(j=1)^N▒〖μ ^_ij F_j 〗)/Y_i +3×(∑_(j=1)^N▒∑_(k=1)^N▒〖μ ^_ik μ ^_kj F_j 〗)/Y_i +

4×(∑_(j=1)^N▒∑_(k=1)^N▒∑_(l=1)^N▒〖μ ^_il μ ^_lk μ ^_kj F_j 〗)/Y_i +⋯ (4)

where, μ ^_ij represents the output of industry i required to produce 1 unit value of industry j’s output (the marker above indicates adjustments for open economies and inventories), F_i represents the portion of the output of industry j that is used for final consumption, and Y_i represents the total output of industry i. Each item on the right side of the equation corresponds to a production link in the industrial chain with varying distances from the final consumption. The numbers to the left of the multiplication sign, i.e., 1,2,3, etc., represent the distance from final final consumption plus one. The part to the left of the multiplication sign in each term represents the portion of the output of industry i that is used in the corresponding position in the industrial chain, as the weight exists. The sum of the items gives the 〖Upstreamness〗_i, it represents the weighted average distance between the output of industry i and final consumption. A higher degree of upstream indicates that the industry is closer to intermediate inputs along the industrial chain, and vice versa, closer to the final product.”

We have not presented the full contents of the revisions here. However, they can be found in the revised paper.

Question 5: The discussion on the research findings is relatively simple and insufficient. I recommend to elaborate the findings focusing on the study context and compare the research results with prior studies in order to manifest the contribution of this study. The discussion section needs significantly more depth. It should include subsections (e.g., theoretical implications, managerial implications, limitations and suggestions for future research). The generalizability of the results should be better discussed.

Reply: Thanks for this valuable comment. We have revised Section 8 according to the reviewer's suggestion (pp.19-22). 

Firstly, we have summarized our research content and methods, and explained our main empirical findings and their theoretical implications in detail. “ In this paper, we examine the links between consumers’ risk perception, market demand and firm innovation, taking advantage of the disclosure and the extensive reporting of the Bawang event in 2010. The theoretical hypothesis section of this paper argues that when a sudden accident occurs, such as a product harm crisis, consumers’ risk perception tends to increase, which, in turn, affects their purchasing behaviour. This change in consumer behaviour leads to a shift in market demand, eventually influencing manufacturing enterprises’ innovation activities. This paper employs a difference-in-differences approach by using the Bawang event as a quasi-natural experiment to identify the impact of changes in consumers’ risk perception on firm innovation. The findings are as follows.

First, consumers’ risk perception effectively promotes firms to innovate. The increase in consumers’ risk perception of products containing dioxane after the Bawang event led to a significant increase in patent applications in the daily chemical and pharmaceutical manufacturing sectors, which rose by approximately 7.6% compared to other manufacturing industries. We propose that consumers’ risk perception affects firm innovation by influencing market demand and provide direct evidence to support this mechanism. In a sudden accident, consumer demand can decrease, leading to a firm’s profits and market share decline. Firms need to take appropriate measures to regain consumer trust and reduce their perception of risk. However, changing factor input or withdrawing from the market can be expensive for firms. Thus, implementing technological or product innovation is the best option for them. 

Second, we examine the impact of the position in the industrial chain and technology spillover on consumers’ perceived risk and firm innovation. Our study has revealed two findings: First, within the same industry chain, the downstream industry is more vulnerable to changes in consumer demand due to its closer proximity to consumers and, therefore, more sensitive to risk perception. This sensitivity makes changes in innovation activities more noticeable in the downstream industry. Second, cross-industry technology spillovers are more likely to occur between firms and industries with high technical similarity when a sudden accident occurs. The higher the technological similarity with the firm involved, the greater the impact of consumers’ risk perception on the innovation of the enterprise.”

Secondly, compared with the existing research, we have provided the possible marginal contribution of our manuscript. (1) The first is the innovation of the research perspective. While there is ample literature on how market demand drives innovation, there is little research on how consumers’ risk perception affects demand-driven innovation. Our research aims to fill this gap and provide a comprehensive understanding of the subject. In addition, our research also examines the strategies that firms can adopt to restore consumer trust and reduce risk perception following product harm crises. While previous studies have mainly focused on external strategies, such as crisis PR and advertising, we find that internal strategies, such as improving product quality and technology through innovation, can also play a crucial role in restoring consumer trust and reducing perceived risks. (2) The second is the innovation of research methods. Previous studies on "demand-driven innovation" have not been successful in solving the endogenous problem and obtaining an unbiased and consistent estimate. However, this paper addresses this issue by selecting the Bawang event as an exogenous shock and using the DID approach method to avoid endogeneity problems. (3) The third is the innovation of theoretical mechanisms. This paper introduces economic concepts such as industrial chain, industrial correlation, and technology spillover effect in the research framework on the impact of consumers’ risk perception on firm innovation. It is the first study that finds a relationship between the influence of consumers’ risk perception on firm innovation and their position in the industrial chain and technical similarity.

Thirdly, based on our research content and empirical results, we have proposed our policy implications from three perspectives: government, firms and consumer. (1) For the government. The study suggests that when the government formulates industrial and innovation policies, it needs to consider the role of the industrial chain and the spillover effect. The government should create matching policy programs for different types of firms. On the one hand, the government should prioritize the development of leading and high-impact industries and increase investment and policy support for them. This approach would promote the common development of other sectors. On the other hand, the government should recognize that some firms may have weak innovation ability and enthusiasm and may struggle to cope with demand-side shocks. To encourage them, the government can stimulate the internal motivation of these firms to innovate through tax incentives, subsidies, and other policies. Additionally, the government should improve the merger and reorganization mechanism of firms. It should eliminate firms with low technical levels and optimize industries through resource integration and factor reconstruction. (2) For the firm. Firms should take advantage of the spillover effect of innovation and participate actively in collaborative innovation activities within their supply chain while optimizing their innovation strategies. Specifically, firms should closely monitor the innovation dynamics of related companies and learn from and absorb their innovation results to improve their product quality and production efficiency, thereby enhancing their competitiveness. Additionally, firms should maintain timely and effective interaction with industry-related firms to keep track of their innovation plans and the development progress of new products. By collaborating with other firms and sharing resources and knowledge, firms can improve their innovation ability and performance. For instance, they can jointly establish joint research and development teams with other firms to develop new technologies and products, reducing costs, shortening research and development cycles, and improving market competitiveness. (3)For the consumers. The government can stimulate the willingness of consumers to consume by providing them with cash subsidies, consumer vouchers, and other incentives. Firms should listen to consumers’ opinions and feedback during production, operation, and research and development and amend their business strategy and research and development direction accordingly. On the other hand, consumers should actively exercise and maintain their right to supervision, right to know, and other relevant rights and collaborate with the government to supervise and give feedback on relevant firms and products. This would encourage firms to provide better products and services to the consumers.

Finally, we have discussed the limitations of our research and the future research directions. During our discussion, we explored the extent to which the results we obtained can be applied or generalized to other contexts. While our empirical findings may have limited applicability, the research concepts and theoretical framework can still be used to investigate similar issues. For instance, we can apply them to study the impact of changes in consumers' perceived risks on market demand and innovation during unexpected events like the COVID-19 pandemic (pp.22-23). In addition, this paper specifically analyzes the changes in risk perception from the perspectives of social risk and functional risk, and it is more detailed than previous studies. However, like previous studies, the discussion on consumers’ risk perception in this paper is still a qualitative analysis. In the future, constructing quantitative indicators of consumers’ risk perception into the empirical model will become a focus of subsequent research.

Question 6: The paper is required for proper writing and needs to be revised with professional academic writing. There are several grammatical issues throughout the paper.

Reply: We apologize for the language issue of our manuscript. We tried our best to improve the manuscript and made some changes to the manuscript. We did not list the changes here but marked them in red in the revised paper. We appreciate for reviewer's warm work earnestly and hope that the correction will meet with approval.

Finally, we hope that the changes we have made solve all your concerns about the article. We are more than happy to make any further changes that will improve the paper. Thank you again for your time and help.

---

## [Decision Letter · Decision Letter 1]

25 Mar 2024

Consumers’ Risk Perception, Market Demand, and Firm Innovation: Evidence from China

PONE-D-23-31451R1

Dear Dr. Ren,

We’re pleased to inform you that your manuscript has been judged scientifically suitable for publication and will be formally accepted for publication once it meets all outstanding technical requirements.

Kind regards,

Academic Editor

PLOS ONE

Additional Editor Comments (optional):

Reviewers' comments:

Reviewer's Responses to Questions

**Comments to the Author**

1. If the authors have adequately addressed your comments raised in a previous round of review and you feel that this manuscript is now acceptable for publication, you may indicate that here to bypass the “Comments to the Author” section, enter your conflict of interest statement in the “Confidential to Editor” section, and submit your "Accept" recommendation.

Reviewer #1: All comments have been addressed

Reviewer #2: (No Response)

2. Is the manuscript technically sound, and do the data support the conclusions?

Reviewer #1: Yes

Reviewer #2: (No Response)

3. Has the statistical analysis been performed appropriately and rigorously? 

Reviewer #1: Yes

Reviewer #2: (No Response)

4. Have the authors made all data underlying the findings in their manuscript fully available?

Reviewer #1: Yes

Reviewer #2: (No Response)

5. Is the manuscript presented in an intelligible fashion and written in standard English?

Reviewer #1: Yes

Reviewer #2: (No Response)

6. Review Comments to the Author

Reviewer #1: I am satisfied with the answers provided by the author. All the corrections are incorporated in the article.

Reviewer #2: Thanks for addressing reviewer concerns. The author addresses all of the reviewer's comments substantive, enhancing the scholarly value of the work

7. PLOS authors have the option to publish the peer review history of their article (what does this mean?). If published, this will include your full peer review and any attached files.

Reviewer #1: No

Reviewer #2: No

---

## [Editor Report · Acceptance letter]

2 Apr 2024

PONE-D-23-31451R1 

PLOS ONE

Dear Dr. Ren, 

I'm pleased to inform you that your manuscript has been deemed suitable for publication in PLOS ONE. Congratulations! Your manuscript is now being handed over to our production team.

Kind regards, 

on behalf of

Dr. Abaid Ullah Zafar 

Academic Editor

PLOS ONE